# Hematopoietic stem-cell gene therapy is associated with restored white matter microvascular function in cerebral adrenoleukodystrophy

Arne Lauer[1,2], Samantha L. Speroni[1,3], Myoung Choi [1], Xiao Da[4], Christine Duncan[5], Siobhan McCarthy[1,3], Vijai Krishnan[1,3], Cole A. Lusk[1,3], David Rohde [6], Mikkel Bo Hansen [7], Jayashree Kalpathy-Cramer [8], Daniel J. Loes[9], Paul A. Caruso[10], David A. Williams[5], Kim Mouridsen[7], Kyrre E. Emblem[11], Florian S. Eichler[1,3] & Patricia L. Musolino [1,3,8] ✉

Blood-brain barrier disruption marks the onset of cerebral adrenoleukodystrophy (CALD), a devastating cerebral demyelinating disease caused by loss of *ABCD1* gene function. The underlying mechanism are not well understood, but evidence suggests that microvascular dysfunction is involved. We analyzed cerebral perfusion imaging in boys with CALD treated with autologous hematopoietic stem-cells transduced with the Lenti-D lentiviral vector that contains *ABCD1* cDNA as part of a single group, open-label phase 2-3 safety and efficacy study (NCT01896102) and patients treated with allogeneic hematopoietic stem cell transplantation. We found widespread and sustained normalization of white matter permeability and microvascular flow. We demonstrate that *ABCD1* functional bone marrow-derived cells can engraft in the cerebral vascular and perivascular space. Inverse correlation between gene dosage and lesion growth suggests that corrected cells contribute long-term to remodeling of brain microvascular function. Further studies are needed to explore the longevity of these effects.

X-linked adrenoleukodystrophy (ALD) is caused by mutations in the *ABCD1* gene, encoding the ALD protein that leads to the accumulation of very long-chain fatty acids (VLCFA) in plasma and tissues[1]. The most severe phenotype is an aggressive form of inflammatory demyelination called cerebral adrenoleukodystrophy (CALD) characterized by focal blood-brain barrier (BBB) disruption and leukocyte infiltration into the white matter (Fig. 1a, b)[2]. If left untreated ~90% of patients with CALD will suffer progressive demyelination resulting in vegetative state or death[3]. The mechanisms that are responsible for the conversion to CALD are not well understood, but may involve a second hit. Human histopathology and perfusion studies suggest that microvascular dysfunction and increased BBB-permeability precede active

[1]Department of Neurology, Massachusetts General Hospital, Boston, MA, USA. [2]Department of Neuroradiology, Heidelberg University, Heidelberg, Germany. [3]Center for Genomic Medicine, Massachusetts General Hospital, Boston, MA, USA. [4]Functional Neuroimaging Laboratory, Department of Psychiatry, Brigham and Women's Hospital, Boston, MA, USA. [5]Dana–Farber and Boston Children's Cancer and Blood Disorders Center and Harvard Medical School, Boston, MA, USA. [6]Center for Systems Biology, Massachusetts General Hospital, Boston, MA, USA. [7]Department of Clinical Medicine, Aarhus University, Aarhus, Denmark. [8]Athinoula A. Martinos Centre for Biomedical Imaging, Charlestown, MA, USA. [9]Suburban Radiologic Consultants, Ltd, Minneapolis, MN, USA. [10]Department of Radiology, Massachusetts General Hospital, Boston, MA, USA. [11]Department of Diagnostic Physics, Oslo University Hospital, Oslo, Norway. ✉e-mail: pmusolino@mgh.harvard.edu

**Fig. 1 | Gene dosage and correlations with demyelinating lesion growth in CALD. a** Representative images of T2-weighted (T2W) and fractional anisotropy (FA) maps of a patient with CALD before (PRE), 1 and 2 years post successful gene therapy (GT). Magnifications illustrate the progress of the T2W lesion and structural tissue reorganization on FA maps primarily within the first year and only minor changes in the second year. **b** Schematic illustrations of microvascular vulnerability caused by *ABCD1* deficiency. Loss of *ABCD1* function in hemizygotes leads to altered interactions of leukocytes and brain-endothelium. Compared to healthy flow conditions (1), this causes increased flow heterogeneity and BBB-permeability within capillary beds (2) and precedes conversion to CALD exacerbated by a yet unknown "second hit" (red). As the CALD manifests, flow heterogeneity and BBB-permeability exacerbate (3). The degree of leukocyte-to-brain-endothelial cell interaction is thought to affect microcirculation causing flow disturbances and shunting in the capillary bed, impairing vascular efficacy (4). **c** Longitudinal data for T2W lesion volume (top) and mean lesional FA (bottom) of GT-treated patients (*n* = 15). For each row left, diagram shows the longitudinal course and right diagram shows relative monthly change in lesion volume and FA PRE to visit 1 (1–2 Mo) and within the first and second-year follow-up (1–12 Mo, 12–24 Mo) post treatment. Blue lines indicate individual, black line represents mean change. Welch's ANOVA between-group difference *P* = 0.0004 and *P* < 0.0001. The *P* values were determined by Dunnett's multiple comparison test. **d** Individual Vector copy number (VCN) in the GT product before infusion (green) and in the peripheral blood (blue) for each patient at follow-up after Infusion (*n* = 11–15, dotted line connects median; error bars indicate interquartile range). **e** Plot showing the relationship of VCN in the GT product and T2W lesion growth over 2 years (*n* = 9, two outliers removed, simple linear regression). **f** Plot showing the relationship of VCN in the GT product and FA decreases over two years (*n* = 11, simple linear regression). **g** Logistic regression curve representing an estimate of the probability of delayed T2W lesion growth depending on VCN in the peripheral blood at 12 months (yes vs. no increase V1-12 months, $\chi^2[1]$ = 7.88, *p* = 0.037, Pseudo $R^2$ = 0.41, Exp(B) = 0.003, 95% CI 0.001–0.707, *n* = 15). Source data are provided as a Source Data file.

demyelination[4,5]. In-vitro studies confirmed that loss of *ABCD1* in human brain microvascular endothelium leads to increased endothelial adhesion and permeability to monocytes[6,7]. Increased leukocyte-endothelial cell interactions can result in slower capillary blood flow and impair vascular efficiency (Fig. 1b)[8]. Indeed, dynamic susceptibility contrast MR (DSC-MR) perfusion studies in ALD patients have shown that *ABCD1* dysfunction alters white matter capillary flow and limits the metabolic rate of oxygen. In addition, regional increases in these flow disturbances precede progression of the active demyelination[5,9–11].

In early disease stages, allogeneic hematopoietic stem-cell transplantation (allo-HSCT) can halt disease progression[12–14]. Recently, gene therapy (GT) with autologous hematopoietic stem cells, which utilizes CD34 + cells transduced ex vivo with a Lenti-D lentiviral vector that contains *ABCD1*-cDNA, has shown promising early results as an alternative to allo-HSCT for children lacking a related donor with an early 88% survival rate[15].

While the exact dosage of gene-corrected cells needed to halt or prevent cerebral disease is unknown, we know that female heterozygotes harboring ~50% of myeloid cells expressing *ABCD1* are protected from developing CALD[16]. Moreover, early GT study results also suggest that lower vector copy numbers (VCN) of the *ABCD1* gene per diploid genome of infused CD34+ cells may negatively impact clinical outcomes[15]. The impact of GT in this disease, like others, may relate to both the transduction efficiency as measured by VCN and the total cell dosage of cells containing the corrected gene expressed as CD34+ cell/

kg. To elucidate plausible mechanisms by which correction of *ABCD1* in HSCs can halt CALD we analyze white matter structural and microvascular changes using advanced MR imaging in a sub-cohort of 15 patients enrolled in the GT STARBEAM study (NCT01896102) at a single center[15].

Here, we show that GT leads to widespread and sustained normalization of white matter permeability and microvascular flow. We provide evidence that bone marrow-derived cells with *ABCD1* can engraft in the cerebral vascular and perivascular space. We propose that improved microvascular flow reflects normalized interactions between corrected circulating leukocytes and endothelial cells, which attenuates the progression of CALD.

## Results and discussion
### Gene dosage and correlations with lesion growth in CALD
Lesion volumetric analysis confirmed that GT effectively decelerates CALD lesion progression and can halt CALD disease in a large proportion of patients within the observation intervals. The most rapid deceleration of lesion growth rate occurred after the first follow-up visit after GT infusion (median days, range; 43.5, 26-183), followed by a less prominent but sustained deceleration over the next two years (Fig. 1c). We investigated lesion growth behavior in a retrospective contemporary cohort treated with standard of care allo-HSCT, which also showed lesion growth deceleration following treatment initiation (Suppl. Fig. 1). While this group had similar baseline lesion scores

prompting rescue treatment (Loes score <10), baseline lesion volumes were significantly larger and observation intervals less frequent (Suppl. Table 1). This limited our ability to directly compare both treatments. GT also stopped the reduction in lesional fractional anisotropy (FA), a diffusion-weighted imaging marker of white matter integrity (Fig. 1c).

Gene therapy resulted in persistent detectable VCN in nucleated cells of the peripheral blood during the two years of observation (Fig. 1d). In patients with higher VCN in drug product there was a trend towards smaller lesion growth and significant retention of lesional FA values (Fig. 1e, f), indicating a gene dose-related response. Baseline lesion variations[17], the interval before treatment, and the immuno-suppressive effects of myeloablative conditioning prior to drug product infusion all have potential confounding effects on lesion growth. In order to reduce these effects, we analyzed relative lesion growth after treatment. A logistic regression model showed that patients with a VCN > 0.63 in peripheral blood at 12 months are more likely to have stable lesion size after the first post-transplant visit (Fig. 1g).

### Effects of allo-HSCT and gene therapy on inflammatory CALD lesions

We then examined BBB disruption as a marker of inflammation in CALD lesions. Currently gadolinium contrast enhancement (CE) on T1-weighted MRI imaging is used to assess BBB-disruption and inflammatory activity in CALD (Fig. 2a). The disappearance of CE, which is thought to indicate a normalization of BBB-permeability and strongly correlates with good neurological outcomes, was seen 60-100 days after allo-HSCT in most patients[13]. In GT-treated patients CE resolution was observed as early as a month after infusion. However, in half of the patients a recurrence of a weak T1-weighted hyperintensity between 6 months and 2 years was reported (Fig. 2b).

Assessment of BBB-permeability using T1-weighted imaging poses challenges as both visual and quantitative measurements are hindered given variability in the timing of image acquisition after contrast

injection and the intrinsic decrease in T1-weighted signal intensity caused by tissue structural changes. To overcome this limitation, BBB abnormalities were quantified from relaxation effects caused by contrast extravasation and calculated as the apparent contrast leakage constant ($K_{app}$) from DSC-MR perfusion data[18]. This parameter is a more sensitive biomarker of BBB-permeability that precedes overt T1-weighted CE in CALD lesions[11]. While lesional $K_{app}$ values significantly decreased compared to pretreatment after GT (Fig. 2c), we found no significant differences between those scans reported to have re-emerging CE vs. CE-negative (Fig. 2d). Two years post-GT > 72% of patients showed resolution of CE on conventional T1-weighted imaging. However, $K_{app}$ values higher than hemizygote ALD patients without CALD (HEM), suggests improvement but not a complete normalization of BBB function. Comparable changes in lesional BBB-permeability were observed in patients treated with allo-HSCT or self-arrested CALD, a rare phenotype (<10%), where lesions halt spontaneously (Fig. 2c, e)[2]. The degree of residual BBB-impairment post-treatment correlated with a decline in FA (Fig. 2f).

Examination of capillary flow heterogeneity (CTH) in CALD perilesional white matter, where increased CTH has been shown to predict lesion progression[11], indicated rapid and sustained normalization of microvascular flow disturbances after GT and allo-HSCT (Fig. 2g). There was a trend towards reduced lesion growth with higher degrees of CTH normalization (Fig. 2h).

In summary, perilesional flow disturbances and lesional BBB-permeability improve rapidly after GT-treatment, indicating successful early attenuation of microvascular dysfunction (Fig. 2a).

### Gene therapy effects on white matter microvascular function

To establish if *ABCD1*-transgene expression in HSC and progeny cells can improve the microcirculatory dysregulation observed beyond the demyelinating lesion, we evaluated interindividual capillary flow disturbances by quantifying the proportional time of tracer retention in

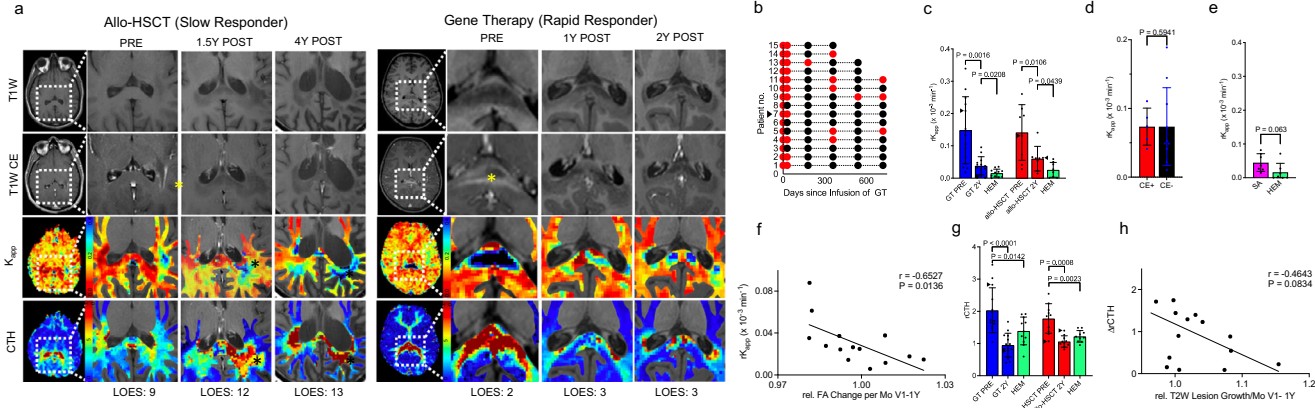

**Fig. 2 | Effects of allo-HSCT and gene therapy on inflammatory CALD lesions.** **a** Representative T1-weighted images without (T1W) and with gadolinium contrast enhancement (T1W CE), the apparent contrast leakage parameter ($K_{app}$) and capillary flow heterogeneity (CTH) maps of two CALD patients before (PRE) and after (POST) treatment with allogenic hematopoietic stem-cell transplantation (allo-HSCT, left) or gene therapy (GT, right). Lesion magnifications in PRE show strong CE ring enhancement which resolved POST treatment in both patients, while POST $K_{app}$- and CTH maps show persisting regionally increased BBB-permeability and microvascular flow heterogeneity in the allo-HSCT patient, in contrast to a GT-treated patient with rapid normalization of microvascular perfusion markers. Data points of these patients are highlighted with pointing triangles in the following figures. **b** Longitudinal data on binary contrast reads (yes vs. no pathologic CE) in GT patients (n = 15). **c** Lesional relative $K_{app}$ PRE and 2Y post GT (n = 13) and allo-HSCT (n = 8) and relative $K_{app}$ in corresponding white matter in age-matched hemizygotes without CALD (HEM, n = 11–8). Error bars

show mean ± SD; the *P* values were determined by two-tailed paired and unpaired students' *t* tests. **d** Comparison of lesional $K_{app}$ in patients assessed positive (CE+) and negative (CE−) for pathologic enhancement at 1Y (n = 5–10). Error bars show mean ± SD; two-tailed unpaired students' *t* tests. **e** Same as **c** for CALD patients with self-arrested lesions (SA, n = 7) vs. age-matched HEM (n = 7). Two-tailed unpaired students' *t* tests. **f** Plot showing the relationship of $K_{app}$ at 1Y and relative monthly FA decrease between the first month POST (V1) and 1Y (n = 15). Line indicates regression; two-tailed Pearson's correlation. **g** Perilesional rCTH PRE and 2Y post GT (n = 13) and allo-HSCT (n = 10) and rCTH in corresponding white matter in age-matched hemizygotes without CALD (HEM, n = 11–10) Error bars show mean ± SD; the *P* values were determined by two-tailed paired and unpaired students' *t* tests. **h** Correlation plot showing perilesional rCTH 1Y and relative monthly T2W lesion growth between the first month POST (V1) and 1Y (n = 15). Line indicates regression; two-tailed Pearson's correlation. Source data are provided as a Source Data file.

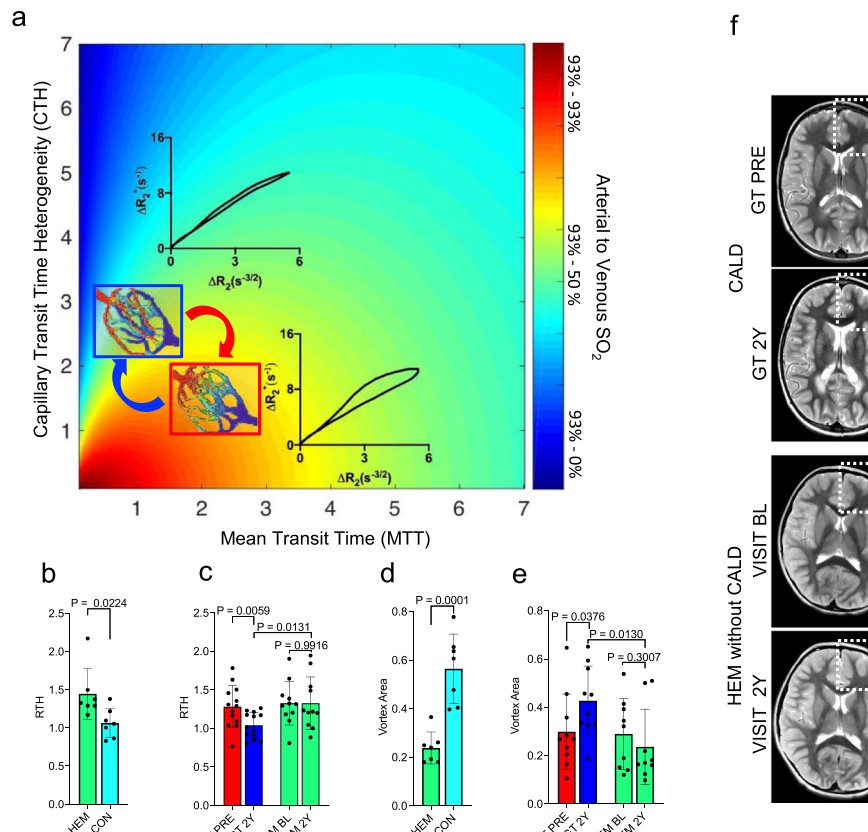

**Fig. 3 | Effects of gene therapy on non-lesional white matter microvascular function. a** Theoretical relationship between the imaging parameters capillary transit time heterogeneity (CTH) and mean transit time (MTT, an overall tissue perfusion marker) regarding the oxygen saturation of the perfused tissue. Flow disturbances in the capillary system lead to a mismatch between MTT and CTH (quantifiable as increased relative transit time heterogeneity = RTH). The microvascular flow delay results in increased oxygen extraction in the capillary bed ($\Delta SO_2$) indicating a reduction in vascular efficacy. This affects the relaxation curves from MRI gradient- and spin-echo signals and forms a reduced relative vortex area (VA). **b** Mean RTH in corresponding NAWM of HEM and controls without ALD (CON, $n = 7$). Data are expressed as mean ± SD; the $P$ values were determined by two-tailed unpaired students' $t$ tests. **c** Mean RTH in distant normal-appearing white

matter (NAWM) in CALD patients ($n = 13$) before (PRE) and 2Y post gene therapy (GT) and baseline to 2Y follow-up visit in age-matched hemizygotes without CALD (HEM, BL vs. 2Y, $n = 11$). Data are expressed as mean ± SD; the $P$ values were determined by two-tailed paired and unpaired students' $t$ tests. **d** Same comparisons as in (**b**) for mean VA ($n = 7$). Data are expressed as mean ± SD; the $P$ values were determined by two-tailed unpaired students' $t$ tests. **e** Same comparisons as in **c** for mean VA ($n = 10-9$). Data are expressed as mean ± SD; the $P$ values were determined by two-tailed paired and unpaired students' $t$ tests. **f** Representative T2-weighted (T2W) images of distant NAWM and corresponding RTH and VA maps in a CALD patient PRE and 2Y post GT and an untreated HEM boy of same age without CALD at baseline and 2 years later. Source data are provided as a Source Data file.

the capillary system relative to its mean transit time, i.e. CTH/MTT, which is referred to as relative transit time heterogeneity (RTH)[10]. Here, an increase above unity is an indicator of microvascular dysregulation as this leads, based on classical flow-diffusion equations, to a reduction in the upper limit of the metabolic rate of oxygen for a given perfusion (Fig. 3a)[19].

The RTH parameter was elevated in white matter of untreated HEM compared to age matched controls (Fig. 3b). Two years after GT for cerebral disease we found a significant decrease of RTH in distant normal appearing white matter (NAWM) but no significant changes in perfusion over time in NAWM of age-matched untreated HEM without cerebral disease (Fig. 3c), suggesting that effects are secondary to treatment and not aging[3,9,18].

We then applied an imaging approach designed to assess microvascular architecture and vascular efficiency by quantifying the shift of shape and peak positions of the relaxation rates from simultaneously acquired gradient- and spin-echo MRI signals forming a vessel vortex area (VA), that is influenced by the vascular efficiency level for each imaging voxel[20]. In theory, an abnormal artery to venule oxygen-extraction corresponds to relative shift in VA (Fig. 3a)[21], which was present in *ABCD1*-deficient HEM (smaller VA) when compared to age-matched non-*ABCD1*-deficient controls (Fig. 3d). Treatment with GT

increased VA, while examination of anatomically corresponding NAWM in untreated HEM again revealed no longitudinal changes in VA (Fig. 3e, f). Similar effects were observed in patients treated with allo-HSCT (Suppl. Fig. 2). Diffusion, estimated vessel density and vessel calibers in the same NAWM over time were not significantly affected, ruling out a confounding microstructural reorganization that could occur as result of aging (Suppl. Fig. 3).

## Brain vasculature engraftment of ABCD1 sufficient cells
Together, our data support the hypothesis that the correction of *ABCD1* expression in HSC can normalize BBB-permeability and capillary flow dynamics extending beyond the CALD lesion (Fig. 4a). Yet it remains unclear if the observed changes in cerebral white matter microcirculation translate into clinical effects. Functional outcome observed in the STARBEAM study appeared to be worse in patients that received a drug product with lower dose (by VCN in drug product). Likely the halting of CALD demyelination, cessation of cerebral inflammation and improved metabolic milieu could be contributing as well. Some of the results suggest a gene dose-dependent effect (measured by VCN and total number of CD34+ cell/kg containing the corrected gene) on examined tissue markers. However, the limited number of patients reduces generalizability and possible confounding

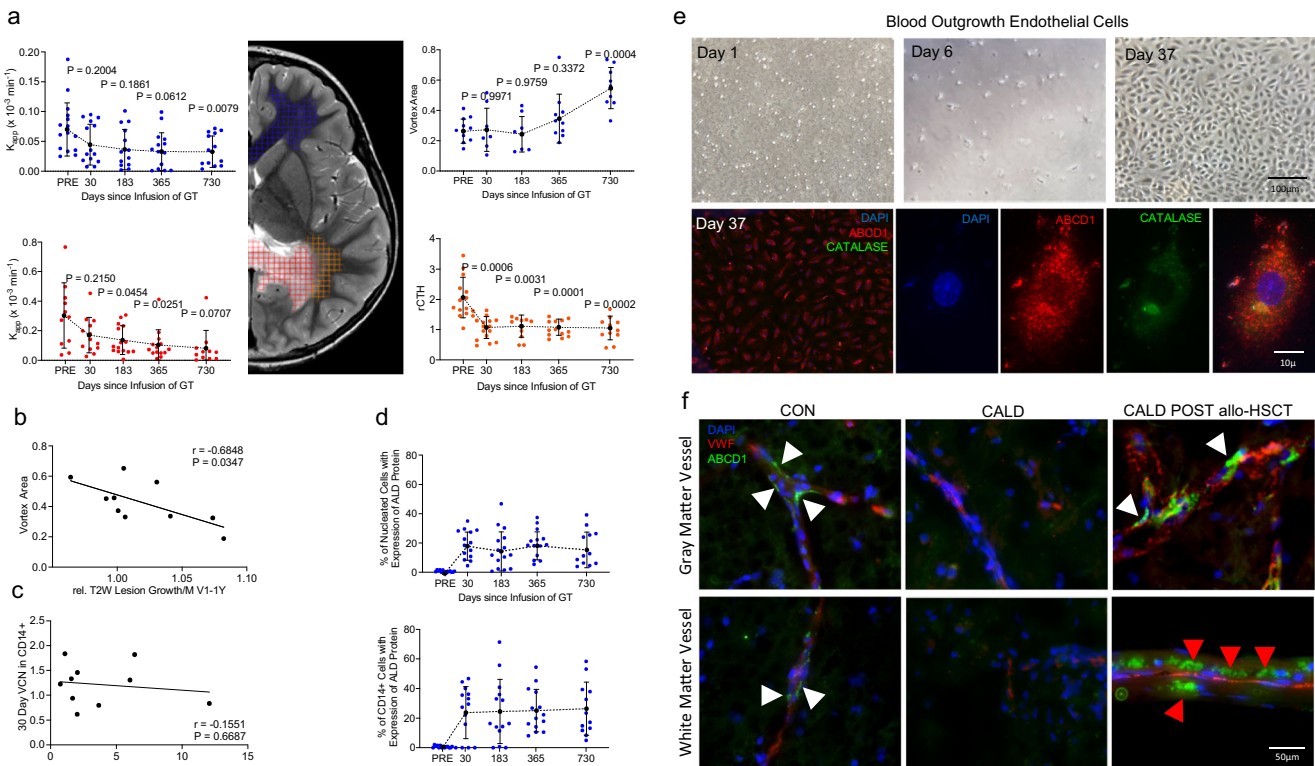

**Fig. 4 | Engraftment of Bone-marrow Derived *ABCD1* Sufficient Cells into the Brain Vasculature. a** Longitudinal mean $K_{app}$ and mean VA in distant NAWM (blue), mean $K_{app}$ in the lesion (red) and mean perilesional rCTH (orange) following gene therapy (GT, $n = 7–15$ each time point). Black dots indicate grouped mean, error bars indicate SD. Mixed effects analysis with Geisser-Greenhouse correction, a single pooled variance and $P$ values (vs. PRE) adjusted for multiplicity. **b** Correlation plot showing post treatment VA in NAWM vs. relative monthly T2w lesion growth V1-1Y ($n = 10$). Line indicates regression; two-tailed Pearson's correlation. **c** Plot showing the relationship of VCN in CD14+ at 30 days and T2W lesion growth over two years ($n = 10$, simple linear regression). **d** Longitudinal data on percentage of CD14+ and nucleated cells expressing ALD protein in the peripheral blood after infusion of GT ($n = 15$). Dotted line connects means; error bars indicate SD. **e** In vitro Blood outgrowth endothelial cell morphology after GT at day 1, 6 and 37 from a patient with at

baseline missing ALDP expressio. Microphotographs of representative confocal imaging show ALDP expression in peroxisomes as demonstrated by colocalization with catalase. Two biological replicates with three technical replicates each were performed. **f** Ex vivo representative microphotographs of brain specimens from a control (CON), an untreated patient with CALD with no *ABCD1* expression, and a patient with CALD treated with allo-HSCT, 15 months after successful bone marrow peripheral blood engraftment. Following allo-HSCT punctate (peroxisomal) high ALDP expression is observed in microvascular pericytes (white arrows) and endothelial cells (white arrowheads) suggesting brain mural vascular engraftment of donor (*ABCD1*+) cells. Large numbers of infiltrating donor-ALDP expressing perivascular mononuclear cells were found in the perilesional white matter micro vessels after allo-HSCT. Three consecutive sections each were processed with the same technique. Source data are provided as a Source Data file.

factors include initial lesion size and progression rate as well as severity of inflammation. Confirmation of such correlations is warranted as it could guide optimization of the gene-drug product. We observed similarities between the GT and allo-HSCT groups regarding effects on lesion growth and white matter microcirculation, but the baseline differences in these groups and our design was not sufficient for non-inferiority studies.

Plausible mechanisms by which replacement of HSC-derived cells expressing *ABCD1* could restore endothelial–monocytes interactions and consequently white matter microvascular dysfunction include (1) direct contribution of bone marrow-derived endothelial precursors to turn over of *ABCD1*-deficient brain microvascular endothelial cells; (2) partial reconstitution of brain mutant macrophage/microglial cells; and/or (3) changes in circulating inflammatory cells and their cytokine profiles. Gene therapy led to a biphasic response with initial rapid lesional changes (<30–60d) and longer-term (>6–12 m) effects including lesional BBB permeability and changes in vascular integrity extending to NAWM (Fig. 4a). The latter correlates with slower lesion growth (Fig. 4b). The lack of correlation between *ABCD1* VCN of CD14+ and growth deceleration (Fig. 4c) taken together with the waning correlation with VCN in the drug product over time (Suppl. Fig. 4) suggests that the initial effects could be attributed to a wave of migration and engraftment of *ABCD1* expressing circulating

progenitor-derived nucleated cells in the peripheral blood (Fig. 4d)[22]. However, the delayed and sustained microvascular flow effects seem to reflect normalized interactions between corrected circulating leukocytes and endothelial cells more likely than, as previously hypothesized, the slow turnover of brain resident cells by progeny of engrafted long-term reconstituting HSCs[23]. Further studies are needed to explore if the percentage of transgene-corrected cells impacts imaging markers or clinical outcome or rather certain thresholds need to be met (as suggested by the extreme rarity of CALD in females heterozygotes with one normal copy of the gene)[16].

Indeed, long-term effects of an HSC correction in CALD have been postulated to be mediated by donor replacement of myeloid-derived cells with *ABCD1* expression, including microglial cells which are thought to originate from HSC-derived monocytes engrafting in the CNS[7,12]. Our findings that the effect upon BBB-permeability and capillary flow last beyond the period of immunosuppression caused by myeloablative chemotherapy (applied 10-1 days prior to infusion) suggest that immunosuppression and cytokine normalization are not the main mechanism underlying lesion growth deceleration and may explain why rapid disease progression occurs when donor HSC failed to engraft despite ablative immunosuppression[13]. Further, the effects extend spatially to areas not affected by inflammatory demyelination.

To determine if GT-associated cells could contribute to microvascular cell precursors, in addition to myeloid (monocyte/microglial cells), we first assessed if CD34+-derived peripheral blood cells could give rise to outgrowth endothelial cells (BOEC) before and after GT from a patient with a mutation lacking ALDP expression at baseline[24]. We found endothelial precursors colonies and 5 weeks later a mature monolayer of characteristic differentiated endothelial cells expressing ALDP in peroxisomes (Fig. 4e). Notably, while only a fraction of leukocytes and monocytes derived from CD34+ corrected cells expressed ALDP in this patient at 24 months (12.7% and 23.4% respectively), 100% of surviving differentiated endothelial cells showed ALDP expression, strongly suggesting that uncorrected precursors with impaired cell cycle regulation and mitochondrial function caused by loss of *ABCD1* function[6,25] are less viable or able to differentiate during in vitro conditions.

While no brain tissue is available in any of the patients treated with GT, we then studied a brain autopsy specimen of a boy with CALD who succumbed to advanced disease 15 months following allo-HSCT. This sample demonstrated for the first time that in a patient with ALD bone marrow-derived cells can engraft long-term in the vascular and perivascular space of white and grey matter. Our immunohistochemistry confirmed that this patient had no detectable levels of ALDP in resident brain cells that normally express the protein endogenously (such as astrocytes, oligodendrocytes, and resting microglia[26]) but striking characteristic punctate peroxisomal ALDP expression was found in circulating and perivascular monocytic cells, endothelium and pericytes at the demyelinating lesion edge and also in the surrounding cortex (Fig. 4f). These data suggest that in addition to correction of the monocyte/macrophage lineage, which could normalize the adherence to microvascular endothelial cells, hematopoetic stem cells could also be providing vascular elements to the cerebral microvasculature that may account for changes in the microvascular function of NAWM on MRI over time ($K_{app}$, CTH, RTH, Vortex Area) following HSCT. As a limitation, the current study does not allow for quantification of the effect of restoration of ABCD1 function in brain endothelium, pericytes, or microglia and their relative contributions to inflammation in CALD. Further research is warranted as the identification of key cells/factors and their temporal interaction in this inflammatory process may lead to the development of new treatment approaches and improved monitoring of CALD.

To summarize, our data show that treatment with autologous hematopoietic stem cells transduced with the Lenti-D lentiviral vector that contains *ABCD1* cDNA halts CALD lesion progression and improves microvascular circulation of the entire cerebral white matter. We found an *ABCD1*-transgene-dependent effect on arresting lesion growth. A small cohort and the lack of longer-term observation warrant caution and further studies are needed to explore the longevity of these effects.

## Methods

### Patients

Fifteen patients, enrolled since October 2013 at the Massachusetts General Hospital (MGH) were recruited as a sub-cohort from the STARBEAM study (ALD-102, NCT01896102), an ongoing multicenter, single group, open-label, phase 2–3 study of gene therapy (GT), which involves infusion of autologous CD34+ hematopoietic stem cells, transduced ex vivo with the elivaldogene tavalentivec lentiviral vector (Lenti-D) that contains ABCD1-cDNA, also known as elivaldogene autotemcel (eli-cel)[15]. Primary outcomes of the trial are percentage of participants who were alive and have no major functional disabilities at Month 24 and are without allo-HSCT or rescue cell administration and the proportion of participants who had experienced either acute or chronic graft versus host disease (GVHD) by Month 24. Secondary outcomes include, among others, percentage of participants who demonstrated resolution of gadolinium positivity on magnetic resonance Imaging at Month 24 (A full list of all secondary outcomes can be found here: https://clinicaltrials.gov/ct2/show/NCT01896102). At the time this manuscript was drafted, 32 patients had been enrolled in the ongoing trial (of which 15 were treated at MGH). Two deaths had occurred since trial initiation (none in patients treated at MGH).

In addition, male ALD patients with CALD lesion size matching inclusion criteria for rescue treatment (Loes score ≤10) and available advanced MRI imaging (dynamic susceptibility contrast magnetic resonance perfusion imaging [DSC-MR-Perfusion] was mandatory for inclusion) were drawn from an ongoing observational cohort of 93 male patients with ALD (including CALD, self-arrested CALD [SA] and hemizygotes without signs of CALD [HEM]) at MGH between January 2006 and October 2019 (Suppl. Table 1). In patients outside of the ALD-102 trial MR imaging was performed for clinical monitoring of disease progression. We also included 7 age-matched patients without *ABCD1*-deficiency as controls, that underwent DSC-MR-Perfusion imaging during routine clinical care and showed no evidence of cerebral perfusion, diffusion, or structural abnormalities, identified by a keyword search[11]. A database query from the NICHD Brain and Tissue Bank for Developmental Disorders was performed. Brain tissue samples from a male CALD patient (10 years of age) who died from the advanced disease after receiving allogeneic hematopoietic stem cell transplantation (allo-HSCT) and from an age-matched untreated CALD patient were obtained for analysis. Informed consent was obtained from all the participants and/or from legal representatives. The study received ethical approval by the Institutional Review Board of MGH (MGH protocol 2012-P-000132/1).

### Treatment and laboratory assessments

Patients were either treated with GT or allogeneic hematopoietic stem cell transplantation (allo-HSCT) as standard of care. Cell transduction was performed in accordance with Good Manufacturing Practice conditions with the use of uniform and validated standard operating procedures. Patients received conditioning with busulfan and cyclophosphamide after which the gene therapy product was infused. Weight-based dosing of busulfan IV will be administered on Days −10, −9, −8, and −7 and cyclophosphamide IV (50 mg/kg/day or adjusted dose for obese subjects) will be administered on Days −5, −4, −3, and −2[15].

Patients treated with allo-HSCT received conditioning with busulfan and cyclophosphamide or busulfan and cyclophosphamide (same length of treatment as in GT) with addition of anti-thymocyte globulin or fludarabine if cord blood was the stem-cell source.

The *ABCD1* vector copy number for diploid genome of CD34+ cells in the GT study was determined by quantitative polymerase-chain-reaction (PCR) assay. Expression of ALD protein was quantified from peripheral-blood mononuclear cells and the CD14+ cell fraction with the use of flow cytometry[15].

For blood outgrowth endothelial cell morphology blood from a patient after treatment with GT was collected and separated with a Ficoll density gradient. The resulting cells were plated on Collagen Type I in six-well plates. EGM2 media with 20% fetal bovine serum was added to the cells and changed every 2 days for 6 weeks. Cells began to differentiate into endothelial cells after two weeks in culture and formed fully confluent endothelial colonies after five weeks. Cells were stained for ALDP and Catalase using anti-ALDP Monoclonal - (clone 2AL-1D6) specific to amino acids 279–482 within the human ALDP protein (Chemicon, MAB2164, 1:500), and goat polyclonal Catalase (Abcam, ab50434, 1:100) primary antibodies at 4 °C overnight, followed by incubation with Goat Anti-Rabbit IgG H&L (Alexa Fluor® 488, 1:1000) conjugated secondary antibodies, and then mounted in mounting medium with DAPI (ProLong® Gold Antifade Reagent with DAPI). The cells were then imaged via immunofluorescence to detect ALDP and Catalase activity using 80i Eclipse Nikon fluorescence and Zeiss confocal microscopes. Two biological replicates with three technical replicates each were performed.

Human brain samples came sectioned and paraffinized from the tissue bank. Sections were deparaffinized, placed in PBS rinse for three washes (5 minutes each), and blocking agent was applied. Incubation was followed at room temperature for 45 minutes in 5% BSA in PBS with 3% normal goat serum (Vector Lab, #S-1000). Primary antibodies (Mouse ABCD1, Origene, TA803208, 1:250, Rabbit (alternating): IBA1, Wako, 019-19741, 1:200 and Von Willebrand Factor, Abcam, ab6994, 1:500) were applied in 3% BSA solution in PBS, and incubated o/n at 4 degrees. Primary antibodies were removed and samples rinsed three times (5 minutes each) in PBS. Secondary antibodies (Goat [mouse]−1 IgG (H + L), Alexa Fluor® 488 conjugate, A-11001, 1:500; Goat [rabbit]−1 IgG (H + L), Alexa Fluor® 555 conjugate, ab150078, 1:500) were applied followed by incubation for 90 minutes at room temperature. Samples were then rinsed three times (5 minutes each) in PBS and DAPI (Pro-Long® Gold Antifade Reagent with DAPI) was applied. Sections were imaged using 80i Eclipse Nikon fluorescence and Zeiss confocal microscopes. Three consecutive sections were processed with the same technique. Controls for non-specific binding of secondary antibody were performed by incubation of a sample and control sections with antibody dilution buffer without the primary antibody. To determine the level of tissue autofluorescence sections were incubated with primary antibodies but no secondary antibody was added to control and sample sections.

## Structural magnetic resonance imaging

The magnetic resonance studies were performed on three magnetic resonance scanners 1.5 T GE Signa HDx (GE healthcare with eight channel head coil), 3.0 T MAGNETOM TioTrim (Siemens; twelve channel head coil), and 3.0 T Prisma fit (Siemens, eighteen channel head coil). The magnetic resonance protocol included axial T2-weighted (T2W; repetition time [TR]/echo time [TE]/slice thickness [ST], matrix, field of view [FOV] in 1.5 T: 5350–7850 ms/95–112 ms/3–4 mm/512 × 512/22 × 22–24 × 24cm and in 3 T: 4200–9000 ms/97–143 ms/3–4 mm/480–512 × 512/22 × 22–24 × 24cm), pre- and post-contrast axial T1-weighted (T1W; TR/TE/ST/matrix/FOV in 1.5 T: 450–600 ms/14–20 ms/3–4 mm/256 × 256/22 × 22–24 × 24cm; in 3.0 T: 500–742 ms/1.74–9.1 ms/3.0 mm/480–512 × 512/24 × 24cm) and axial diffusion tensor imaging (DTI; TR/TE/ST/matrix/FOV/number of directions in 1.5 T: 10000 ms/87–99 ms/2.2–2.4 mm/256 × 256/22 × 22–cm/25; and in 3 T: 3800–9700–ms/90–92 ms/2.0–5.0 mm/160 × 160/23 × 23 cm/25). Younger children and symptomatic subjects required general anesthesia, while asymptomatic subjects older than 6 years underwent magnetic resonance scanning without sedation.

Dynamic susceptibility contrast magnetic resonance perfusion imaging (DSC MR) was performed for all subjects using gradient echo (GRE) echo planar imaging (EPI) sequences. Acquisition parameters were TR/ST/matrix/FOV: 1500 ms/40 ms/5 mm/128 × 128/22 × 22 cm for 1.5 T and both 3 T scanners. The TE was 40 ms, 32 ms, and 35 ms at 1.5 T and at 3 T scanners. Further, only with the 3 T MAGNETOM Tio-Trim system additional spin-echo (SE) DSC perfusion image series were acquired simultaneously to the GRE EPI sequences with a TE of 96 ms and otherwise identical parameters (Siemens dual-echo planar image protocol).

A total of 60 (1.5 T) or 80 (3.0 T) dynamic acquisitions were acquired before, during, and after injection of 0.1 mM/kg Gadolinium-based contrast. Contrast was injected at 5 ml/s for older subjects using a power injector, while the injection was performed manually for the pediatric subjects using maximal permissible rates depending on their intravenous access (up to 5 ml/s). Incomplete magnetic resonance perfusion datasets or those significantly degraded by motion artifacts were excluded from the analysis.

## Image processing

From the diffusion tensor data, voxel-by-voxel brain maps (256 × 256 matrix) of the isotropic apparent diffusion co-efficient (ADC) and

fractional anisotropy (FA) were generated. FA maps were used to evaluate the integrity of white matter within the lesion slightly modified from previously described to fit selected regions of interest[27].

The included DSC MR datasets were post-processed using PEN-GUIN perfusion software version 1.0 (www.cfin.au.dk/software/penguin). Perfusion maps were calculated using deconvolution with an automatically selected arterial input function. Capillary transit time heterogeneity (CTH) was estimated as the standard deviation of a model transit time distribution obtained as a part of voxel-wise fitting of a vascular model to individual concentration-time curves obtained from the DSC MRI data[10,18]. The relative transit time heterogeneity (RTH) was calculated as CTH divided by the mean transit time (MTT) calculated as the expected value of the transit time distribution estimated adaptively during the voxel-wise fitting similar to CTH. Denoting the adaptively estimated transit time distribution $h(t)$, MTT and CTH are formally found as:

$$MTT = \int_0^\infty t h(t) dt \tag{1}$$

$$CTH^2 = \int_0^\infty (t - MTT)^2 h(t) dt = \int_0^\infty t^2 h(t) dt - MTT^2 \tag{2}$$

whereas RTH is simply found as a fraction of MTT/CTH, i.e.,

$$RTH = \frac{CTH}{MTT} \tag{3}$$

extravasation of contrast agent was quantified to generate the BBB-permeability ($K_{app}$) maps from the tissue residue function in DSC MR Perfusion data slightly modified from[28].

Vessel architectural imaging analysis was performed using custom-made software in MATLAB[20]. Briefly, relaxation rate curves were obtained from the DSC images of the GRE and SE sequences, leakage-corrected, and fitted to a gamma-variate curve. Cerebral blood volume maps were created by standard singular value decomposition with automatically detected AIF and corrected for contrast agent extravasation. Each corresponding voxel-level parametric plot of the pair-wise GRE- and SE temporal data points forms a vortex, from which the vortex area (VA) is calculated directly[21].

All images were converted into the NIfTI image format. Perfusion and diffusion-based maps were co-registered to structural MRI images using an attribute-based image registration algorithm (DRAMMS v1.5.1, https://github.com/ouyangming/DRAMMS and 3D-SLICER v4.6.2, http://www.slicer.org).

## Lesion segmentation and regions of interest

The 3D-SLICER software was used to place regions of interest (ROI) and generate mean output for co-registered maps. In patients with CALD, different types of ROI based on T2W structural imaging were selected. First, to evaluate lesion size, structural integrity and BBB-permeability within the lesion the volumetric CALD lesion defined as hyperintense white matter on T2W scans was delineated using a semi-automated protocol[29]. In the first step the threshold was set to exclude normal-appearing white matter (NAWM), in the second step CALD lesions were manually highlighted and volumes were calculated within the same software. Second, to evaluate microvascular perfusion related to the inflammatory lesion bilateral ROI (equivalent to 24 voxels on perfusion maps) were placed in NAWM directly adjacent to the lesion. Perfusion abnormalities in this ROI have been shown to precede lesion progression[11]. Third, in all patients, including those without lesions (HEM) nine additional ROI of identical size were placed bilaterally within the cerebral white matter (splenium of the corpus callosum, major forceps; occipital white matter; posterior periventricular white

matter; internal capsule white matter; frontal periventricular white matter; lesser forceps; genu of the corpus callosum and frontal white matter) for normalization purposes[11] and to calculate average perfusion values for anatomical corresponding white matter in age-matched HEM. Fourth, to analyze longitudinal therapeutic effects on NAWM, large bilateral ROI (equivalent to 72 voxels on perfusion maps) were placed in NAWM remote to CALD lesions (frontal WM if CALD lesion is located posterior and vice versa).

For every scan ROI placement was performed blinded to perfusion and clinical data (with the exception of apparent CALD lesions on T2W imaging) by two independent readers (intra-class correlation coefficient ranged from 0.86 for T2W lesion volumes to 0.84 for perfusion imaging-based parameters). Scans of patients with CALD lesions (GT, allo-HSCT, SA) were age-matched to scans of ALD patients without CALD lesions by nearest neighbor matching (R package "MatchIt v4.5.0", method = "optimal") for specific analysis (differences in means [age in years ± SD] after matching for HEM - GT/allo-HSCT/SA = −0.7367 ±0.99/2.482 ± 2.4/8.04 ± 6.07)[30].

### Evaluation of structural magnetic resonance imaging

Subjects treated with GT were reviewed prospectively or gadolinium contrast enhancement (CE) status during the trial by a pediatric neuroradiologist as central reader (D.J.L.). In all other patients CE status was evaluated by an experienced pediatric neuroradiologist (P.C.) and two pediatric neurologists (P.L.M and F.S.E.). The absence of CALD was determined if baseline and next follow-up visit in male ALD patients (hemizygotes, HEM) showed no cerebral lesions. Untreated ALD patients presenting with a CALD-consistent lesion without contrast enhancement on T1W maps and without lesion progression in the observation period were defined as self-arrested (SA). The investigators who performed the MRI scoring were blinded to the results of the magnetic resonance perfusion but were aware of the treatment status.

### Statistical analysis

By using the means and common standard deviation for ADC, CTH, $K_{app}$, RTH and VA from pilot data and assuming a two-sided alpha-level of 0.05, homogeneous variances for the samples to be compared and a 80.0% power we calculated that small effect sizes can be detected with lesion size and diffusion tensor data (Hedges' $g = 0.11$) and large effect sizes with perfusion based data (Hedges' g = 1.04-1.25) for the numbers of available patients, respectively. In subjects with cerebral disease monthly lesion growth rate and FA decline relative to the previous visit was calculated. In the GT group lesion growth- and FA-decline rates between PRE to the first post treatment visit (V1, 1-2 months) and following first and second-year visits were compared. In the allo-HSCT group lesion growth rates in the first- and second-year POST treatment were compared because early post treatment visits (1–2 months) were not routinely performed. Shapiro-Wilk test was used to test for normality. Longitudinal lesion growth- and FA-decline rates were compared using Welch's ANOVA and post-hoc Dunnetts correction. Simple linear regression analysis was used to investigate associations between VCN in the GT product and imaging markers of lesion progression. The iterative Grubbs method (Alpha = 0.05) was used to determine and exclude outliers before data was entered into the simple linear regression models. A binary logistic regression was used to evaluate the probability of post-transplant lesion progression (defined as positive longitudinal change in T2W lesion size) and the numbers of VCN per diploid genome at 12 months post-transplant. Mean perfusion (CTH, $K_{app}$) within and adjacent to the apparent CALD lesion (or in corresponding white matter if CALD was absent) relative to distant NAWM were evaluated using one-way ANOVA with repeated measures followed by a Dunnett's multiple comparisons test. Paired Student's $t$ test was used to compare longitudinal perfusion values (RTH, VA)

within groups and unpaired Student's $t$ test was used to compare post treatment patients to HEM. To account for repeated measurements and missing data a mixed effects model with treatment as a fixed effect and individual patients as random effects was used to compare longitudinal changes of imaging markers ($K_{app}$, VA and rCTH). Two-tailed Mann–Whitney tests were used to compare baseline CALD lesion volumes, Loes Scores, follow up visits and time, unpaired students' t-tests to compare mean lesional FA and Fisher's Exact test to compare percentage of dual echo perfusion protocols GT vs. allo-HSCT. Statistical analysis was performed using SPSS v22.0 (IBM Corp. Armonk, NY), Graphpad Prism v9.5.0 (Graphpad Software, San Diego, CA), and R v3.5.2 (cran.r-project.org).

### Reporting summary

Further information on research design is available in the Nature Portfolio Reporting Summary linked to this article.

## Data availability

STARBEAM study-related data that supports the findings of this study are available from Bluebird Bio. Restrictions apply to the availability of these data because elements of the data set comprise information proprietary to Bluebird Bio. Any requests for additional data will be considered by all authors and Bluebird Bio. Bluebird requires 30 days from the receipt date of the request to consider and respond to data requests. Any requests can be sent to medinfo@bluebirdbio.com. Data not associated with the STARBEAM study and collected at MGH can be shared with interested investigators but are subject to local and national ethics regulations and legal requirements that respect the informed consent forms. The raw imaging data are protected and are not available due to data privacy laws. Source data are provided in this paper.

## Code availability

The custom code and mathematical algorithms are available upon request. Please contact K.E.M. (kemblem@ous-hf.no) for vessel architectural imaging. Please contact M.B.H. (mbh@cercare-medical.com) for background on the contrast agent leakage calculations and capillary transit time heterogeneity measurements. A clinical version of the perfusion algorithms used is available through Cercare Medical, please contact ha@cercare-medical.com for inquiries on the use of contrast agent leakage and capillary transit time heterogeneity-enabled perfusion processing in clinical settings.

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

## Acknowledgements

We are thankful to the families and their respective local care providers for their participation in this study. K.E. receives research support from the European Research Council grant 758657 ('ImPRESS'); Southern and Eastern Norway Regional Health Authority grants 2017073 and 2013069; The Research Council of Norway grant 261984. F.S.E. receives research support from NINDS (U54 Global Leukodystrophy Initiative Clinical Trials Network (GLIA-CTN)]), the European Leukodystrophy Association, the Leblang Foundation, the Arrivederci Foundation, and the Hammer Family Fund. P.L.M. receives research support from NINDS (K08 NS094683-01), Child Neurology Foundation, MGH Department of Neurology, Minoryx Therapeutics, and Dooley's Family Fund.

## Author contributions

A.L.: study concept and design, analysis, and interpretation of data, and drafting of the manuscript. S.L.S.: data collection, analysis and interpretation of data, and drafting of the manuscript. M.C.: data collection and revising of the manuscript. X.D.: data collection, analysis and interpretation of data, and drafting of the manuscript. C.D.: data collection and revising of the manuscript. S.M.C.: data collection and revising of the manuscript. V.K.: data collection and revising of the manuscript. C.A.L.: data collection and revising of the manuscript. D.R.: Data collection and revising of the manuscript. M.B.H.: the creation of new software used in this work, analysis, and interpretation of data, critical revision of the manuscript for important intellectual content. J.K.-C.: Critical revision of the manuscript for important intellectual content. D.J.L.: data collection and revision of the manuscript. P.C: Data collection and revision of the manuscript. D.W.: Data collection and revision of the manuscript. K.M.: the creation of new software used in this work, critical revision of the manuscript for important intellectual content. K.E.: the creation of new software used in this work, analysis, and interpretation of data, critical revision of the manuscript for important intellectual content. F.S.E.: study concept and design, analysis, and interpretation of data, drafting of the manuscript, and study supervision. P.L.M.: study concept and design, analysis, and interpretation of data, drafting of the manuscript, and study supervision.

## Competing interests

This work was funded primarily by NINDS K08 NS094683-01 and R01NS117575 with a partial contribution from Bluebird Bio. D.J.L. is a consultant for Bluebird Bio. D.W. has received research funding for research in hemoglobinopathies and licensed certain IP relevant to hemoglobinopathies to Bluebird Bio. He has received payments from Bluebird Bio in past through BCH institutional licensing agreement and has the potential for future royalty/milestone income. P.L.M. is the co-I of Bluebird Bio clinical trials. She is a consultant to bluebird bio. F.S.E. is the co-PI of Bluebird Bio clinical trials. M.B.H and K.M. are co-applicants on a patent application based on the presented techniques in this manuscript (PCT/DK2014/050296) and are shareholders in Cercare Medical ApS. A.L.; S.L.S.; M.C.; X.D.; C.D.; S.M.C.; V.K.; C.A.L.; D.R.; J.K.-C.; K.E., and P.C. do not have any disclosures to report directly relating to this study.
