## [Peer Review File · Nature Communications]

Hematopoietic stem-cell gene therapy is associated with restored white matter microvascular function in cerebral adrenoleukodystrophyREVIEWER COMMENTS

Reviewer #1 (Remarks to the Author):

In the manuscript "Gene Correction in Hematopoietic Stem Cells Restores White Matter Microvascular Function in Cerebral Adrenoleukodystrophy" by Lauer et al convincingly demonstrated that GT restores white matter microvascular function in CALD. The correction is long lasting and affected also the microvascular function in non-lesional white matter. The correlation between ABCD1 gene dosage and the demyelinating lesion growth was investigated. The gadolinium contrast enhancement in T1-weighted images is compared to capillary flow heterogeneity maps before and post ABCD1 gene therapy in a longitudinal follow up.

There are some comments:

1. In Fig. 1 b it is not clear what exactly was the status of the lesion shown in 2. Was the X-ALD patient at the time of imaging age-matched hemizygote X-ALD patient without detectable demyelination?
2. In Fig. 1d according to the figure legend the blue dots indicate the vector copy numbers per diploid genome after GT in the 11-15 patients per time point. The dotted line connects the means. Given the distribution of the blue dots the dotted line seems unlikely to connect the means (at least it looks like almost all dots are below the mean). In any case if vector copy numbers per diploid genomes are not normally distributed the median might be more appropriate?
3. Figure 1e: What is the reason for removing the outliers? Why are they charged as outliers? As the correlation is very weak this issue should be explained in more detail by explaining why two samples had to be removed before analysing.
4. Fig 2a: Was the overall pre-treatment status of the inflammatory lesion comparable between the patients undergoing HSCT and that undergoing GT depicted as representative images cases e.g. in LOES score? This should be mentioned in the manuscript. It would be nice if a star (or coloured dot or arrow), instead of a dot could indicate in Fig 2c which patient has been shown as representative in 2a.
5. Fig 3b. In order to appreciate the effect on the RTH after GT in NAWM as well as the alterations of RTH in non-inflammatory X-ALD patients it would be highly appreciated if the RTH data could be provided for the control cohort (as it is shown for the Vortex Area in panel c).
6. Fig 3b and 3e: The nomenclature could be changed: GT PRE, GT 2Y, HEM BI, HEM 2Y.
7. Discussion line 136: "...capillary flow dynamics in a dose-dependent manner (measured by VCN and total number of CD34+ cell/kg containing the corrected gene)." This is a very strong statement given the weak correlation in fig 1e. Due to the small number of samples, the initial degree of inflammation or lesion progression could also coincidentally contribute to the observed result. Thus, this should be discussed as a possible limitation.
8. In line 299 it is described that also the CD14+ cells have been isolated. It would be of major interest if the CD14+ vector copy numbers per diploid genomes at e.g. 30 days post GT correlates stronger with T2W lesion growth over two years (similar to Fig. 1e) and FA decrease over two years (similar to Fig. 1f).
9. Discussion line 140: "...1) direct correction of ABCD1-deficient brain microvascular endothelial cells." Does that mean that the authors suspect that the brain microvascular endothelial cells can be exchanged by GT derived cells? The partial reconstitution of brain mutant macrophage/microglial cells and changes in circulating inflammatory cells and their cytokine profiles appear most likely. The correction of monocyte/macrophage lineage by GT would normalize the adherence to microvascular endothelial cells. This would be in the best agreement with the overall long term correction of the even

in the NAWM in comparison to the hemizygote X-ALD patient without inflammation. Within the early phase very likely partial reconstitution of brain mutant macrophage/microglial cells as well as the overall change of inflammatory status are of importance for the observed findings. A global direct correction of ABCD1-deficient brain microvascular endothelial cells however would require much more explanation and references in order to be understood.

Reviewer #2 (Remarks to the Author):

Thank you for the opportunity to review this novel paper from a very able group in this field. Many have been eagerly waiting for more reports of findings related to the GT for this disease. Overall this paper offers very important new insight into evaluating the pathological processes that occur in cerebral ALD, and the authors provide a critical focus on microvascular function, given what we know about BBB disruption in this devastating disease. I have some questions for the group.

1) I very much appreciate that the authors put the GT findings in context by, at points, comparing to allo-HSCT outcomes. Yet the groups are different by initial lesion severity, with the allo-HSCT group having more severe initial lesions than the GT group. Specifically, while it says that lesion sizes matched between the groups, the supple table 1 suggests otherwise? Initial (i.e., pre-treatment) lesion "size" has been an important predictor of outcomes, including the behavior of the lesion following allo-HSCT. For example, Pierpont and group (Neurology 2020) recently reported that more severe lesions before allo-HSCT tended to increase in severity at a more rapid pace after HSCT; in contrast less severe lesions increased in severity at a less rapid pace after HSCT. This makes me wonder:

a. Given that the allo-HSCT group has more severe initial lesions, which would predict worse growth behavior, should our hopes for GT be muted by the fact that this less severe GT group is showing "Similar lesion growth behavior" (lines 73-74) to a more severe group (allo)? I would appreciate that the authors somehow represent this because to say that the two therapies show similarities might be too reassuring.

b. Knowing that pre-treatment lesion severity has been identified as important by other groups, I do not find to be adequate this justification for removing this factor and instead focusing on the behavior of lesions beginning with post-treatment: "To eliminate the effects of baseline lesion variations," as this author group cites their prior work. Perhaps it would reduce the appearance of bias if the authors elaborate upon the scientific rationale for beginning with the lesion severity post treatment and acknowledge others' findings in this regard.

2) Could the authors specify length of time on immunosuppressive treatment? I appreciate that the authors address it and give their assertion that it is unlikely to explain findings. Just wish to know more detail on it.

3) While there are many interesting elements in the figures, they might be a bit busy with so very many panels. Anywhere the authors could be willing to trim? It is leaning toward creating some burden for the reader to sort through the many aspects and map up against the legends and the text.

4) Possibly rewording for clarity lines 54-55, "Focal increases in these flow disturbances signal where active demyelination will later ensue." Given the wide readership of this journal, I think the less familiar audience may misinterpret this sentence to mean that there are accepted, reliable ways to determine which boys will convert to the cerebral form.

Reviewer #3 (Remarks to the Author):

Lauer and the colleague investigated white matter microvascular function based on MRI findings.

The authors described that they analyzed "a sub-cohort of 15 patients". According to the original report (NEJM 2017), 17 patients were enrolled for the gene therapy (GT), and two of them died during the course. The author should explicitly describe this point,

The authors described that Lesion volumetric analysis confirmed that GT effectively arrests CALD disease progression. In the original report (NEJM 2017), lesion progression, as measured with the Loes score, had stabilized in 12 of the 17 patients. The authors need to be careful to describe the results. There seems to be a bias in the description.

The authors described that a rapid deceleration of lesion growth rate occurred as early as 1 month after GT initiation. How can such a conclusive statement be made within one month? The reviewer does not think it is possible.

The authors described that their data support the hypothesis that the correction of ABCD1 expression in HSC can normalize BBB-permeability and capillary flow dynamics in a dose-dependent manner. The observation of changes in BBB-permeability is interesting. Similar findings have also been described in the patients who underwent HSCT. It is unclear, however, whether ABCD1 expression in HSC directly changed the functions of the endothelial cells. It may well be a secondary result. The authors need careful consideration on this point.

The authors suggest three mechanisms of the effect of GT on BB-permeability, including 1. direct correction of ABCD1-deficient brain microvascular endothelial cells, 2. partial reconstitution of brain mutant macrophage/microglial cells, and/or 3. changes in circulating inflammatory cells and their cytokine profiles. "Endothelial-monocytes interactions" is obscure wording. Consider clear expressions. The authors need to explain the rationale for "direct correction of ABCD1-deficient brain microvascular endothelial cells". Is there any evidence that endothelial cells are directly corrected by GT?

Treatment with the Lenti-D HSC GT halts CALD lesion progression and improves microvascular circulation of the entire cerebral white matter. This description is clearly overstated and should be carefully rephrased. In the original report (NEJM 2017), the clinical effect is limited to partial effect in some patients.

It is unclear whether the changes in BBB-permeability are a direct result of changes in endothelial cell functions with GT, or a secondary phenomenon as a result of ABCD1-expressing bone marrow-derived cells. Since this point is essential, the authors need to discuss on this point in detail.

Recent publication of HSCT for adult CALD (Brain Communications, 2020, <https://doi.org/10.1093/braincomms/fcz048>) should be referred.

Dear Reviewer,

We would like to thank the reviewers and editors for their thoughtful comments and constructive suggestions which have significantly improved our manuscript (NCOMMS-20-38677-T) entitled: "Gene Correction in Hematopoietic Stem Cells Restores White Matter Microvascular Function in Cerebral Adrenoleukodystrophy".

To address the concerns and respond to the encouraging and very helpful reviewers comments we spent several months recruiting subjects and performing new *in vitro* and *ex vivo* studies to provide further data on ABCD1 expressing brain engrafted cells following Gene Therapy (GT) and allogenic hematopoietic stem cell transplantation (allo-HSCT). Despite a number of delays caused by the Covid-19 pandemic restrictions we were able to obtain specimens from two subjects that underwent gene therapy and allogenic hematopoietic stem cell transplant to provide evidence of 1) corrected CD34+ cells (Lenti-D Gene Therapy) generating functional brain endothelial precursors; 2) long-term brain engraftment of bone-marrow derived endothelial, pericytes and monocytic perivascular cells; and 3) VCN in GT drug product best predicts lesion growth deceleration supporting an early wave of brain engrafting cells accounting for the largest effect on lesion growth deceleration. We think these findings in conjunction with the perfusion imaging data significantly increase the impact and novelty of our manuscript which now shows for the first time in humans with inflammatory cerebral demyelination that bone marrow-derived cells engraft in brain microvessels and improve blood brain barrier function.

The FDA advisory committee has recently unanimously endorsed eli-cel (ex vivo Lenti-D gene therapy) for CALD. We hope the novel discovery and major revision of our manuscript address the reviewers' comments. Detailed responses to all comments are outlined below. Changes to the manuscript were highlighted with color and summarized point by point:

REVIEWER COMMENTS

Reviewer #1 (Remarks to the Author):

In the manuscript "Gene Correction in Hematopoietic Stem Cells Restores White Matter Microvascular Function in Cerebral Adrenoleukodystrophy" by Lauer et al convincingly demonstrated that GT restores white matter microvascular function in CALD. The correction is long lasting and affected also the microvascular function in non-lesional white matter. The correlation between ABCD1 gene dosage and the demyelinating lesion growth was investigated. The gadolinium contrast enhancement in T1-weighted images is compared to capillary flow heterogeneity maps before and post ABCD1 gene therapy in a longitudinal follow up.

There are some comments:

1. In Fig. 1 b it is not clear what exactly was the status of the lesion shown in 2. Was the X-ALD patient at the time of imaging age-matched hemizygote X-ALD patient without detectable demyelination?

We appreciate this request for clarification. Figure 1B is a schematic illustration of the microvascular vulnerability caused by ABCD1-dysfunction. Assuming a disease model of increased brain-endothelium to leukocyte interaction and increased blood brain barrier permeability, (2) illustrates alterations caused by ABCD1 dysfunction in X-ALD hemizygotes preceding manifest inflammatory cerebral demyelination (absence of CALD) compared to ideal flow conditions seen in (1). In this model (3) illustrates conversion and progression of CALD. We now provide more descriptions in the figure legend:

...b) Schematic illustrations of microvascular vulnerability caused by ABCD1-deficiency. Loss of ABCD1 function in hemizygotes leads to altered interactions of leukocytes and brain-endothelium. Compared

to healthy flow conditions (1), this causes increased flow heterogeneity and BBB-permeability within capillary beds (2) and precedes conversion to CALD exacerbated a yet unknown "second hit" (red). As the CALD manifests, flow heterogeneity and BBB-permeability exacerbate (3). The degree of leukocyte to brain-endothelial cell interaction is thought to affect microcirculation causing flow disturbances and shunting in the capillary bed, impairing vascular efficacy (4)... (p.27)

2. In Fig. 1d according to the figure legend the blue dots indicate the vector copy numbers per diploid genome after GT in the 11-15 patients per time point. The dotted line connects the means. Given the distribution of the blue dots the dotted line seems unlikely to connect the means (at least it looks like almost all dots are below the mean). In any case if vector copy numbers per diploid genomes are not normally distributed the median might be more appropriate?

We thank the reviewer for pointing this out. A scaling mistake is responsible for the upshift of the dotted connecting line. The data indeed did not pass testing for normal distribution. We now provide the corrected graph and have change the figure 1d:

The figure legend was changed accordingly:

d) Individual Vector copy number (VCN) in the GT product (GTP) before infusion (green) and in the peripheral blood (blue) for each patient at follow up after Infusion (n=11-15, dotted line connects the groups median, error bars indicate interquartile range). (p.27)

3. Figure 1e: What is the reason for removing the outliers? Why are they charged as outliers? As the correlation is very weak this issue should be explained in more detail by explaining why two samples had to be removed before analyzing.

We agree that this important detail is missing. Outliers were removed to comply with general assumptions for linear regression analysis. We used the iterative Grubbs method (Alpha = 0.05) to determine outliers and excluded those. This is now added in the Methods / Statistical analysis section of the revised version of the manuscript:

...Simple linear regression analysis was used to investigate associations between VCN in the GT product and imaging markers of lesion progression. The iterative Grubbs method (Alpha = 0.05) was

used to determine and exclude outliers before data was entered into the simple linear regression models... (p.23)

4. Fig 2a: Was the overall pre-treatment status of the inflammatory lesion comparable between the patients undergoing HSCT and that undergoing GT depicted as representative images cases e.g. in LOES score? This should be mentioned in the manuscript. It would be nice if a star (or coloured dot or arrow), instead of a dot could indicate in Fig 2c which patient has been shown as representative in 2a.

One of the limitations of the allo-HSCT cohort are indeed the differences in lesion size, which in addition to the retrospective nature and differences in observation intervals preclude any direct comparisons between the groups. While patients in both groups had early lesions which justified standard of care indication for rescue treatment with allo-HSCT (defined as Loes Score ≤ 10), the lesions in the GT cohort are significantly smaller (as shown in suppl. table 1). The difference is also apparent if Loes' scores are compared (GT vs. allo-HSCT median, [IQR]: 1 (1-2) vs. 5.75 (1-9), $p=0.0214$). We now provide Loes' scores in Figure 2a and suppl. table 1 as well. In addition, we clarified the matching criteria in the methods section and include a short discussion as limitation to our study:

... In addition, male ALD patients with CALD lesion size matching inclusion criteria for rescue treatment (Loes score ≤ 10) and available advanced MRI imaging ... (p.16)

And:

... We investigated lesion growth behavior in a retrospective contemporary cohort treated with standard of care allo-HSCT, which showed expected lesion growth deceleration following treatment initiation (suppl.fig.1). While this group had similar baseline lesion Loes scores prompting rescue treatment (Loes score ≤ 10), baseline lesion volumes were significantly larger and observation intervals less frequent (suppl. tabl. 1). This limited our ability to directly compare both treatments...

(p.4)

We also now highlight the patients depicted in figure 2a with a five-pointed star in figures 2b, c, d and g.

The legends were changed, respectively (p.29).

5. Fig 3b. In order to appreciate the effect on the RTH after GT in NAWM as well as the alterations of RTH in non-inflammatory X-ALD patients it would be highly appreciated if the RTH data could be provided for the control cohort (as it is shown for the Vortex Area in panel c).

Thank you for this request, we now include this data as a new figure 3b and discuss it in the main text:

...b) Mean RTH in corresponding NAWM of HEM and controls without ALD (CON, n=7). Data are expressed as mean+/-SD; *p< 0.05; two-tailed unpaired students t-tests... (p.31)

And:

...The RTH parameter was elevated in white matter of untreated HEM compared to age matched controls (Fig.3b). Two years after GT for cerebral disease we found a significant decrease of RTH in distant normal appearing white matter (NAWM) but no significant changes in perfusion over time in NAWM of age-matched untreated HEM without cerebral disease (Fig.3b), suggesting that effects are secondary to treatment and not aging.^{3,10,17}... (p.6)

6. Fig 3b and 3e: The nomenclature could be changed: GT PRE, GT 2Y, HEM BI, HEM 2Y.

We changed the column labels in Fig. 2c, 2g, 3c, 3e and 3f, accordingly (fig.2-3, p.27-30).

7. Discussion line 136: "...capillary flow dynamics in a dose-dependent manner (measured by VCN and total number of CD34+ cell/kg containing the corrected gene)." This is a very strong statement

giving the weak correlation in fig 1e. Due to the small number of samples, the initial degree of inflammation or lesion progression could also coincidentally contribute to the observed result. Thus, this should be discussed as a possible limitations.

We appreciate the reviewer's comment. The limited number of patients that received GT could result in spurious findings. We carefully rephrased and now include additional discussion to this paragraph:

..Some of the results suggest a gene dose-dependent effect (measured by VCN and total number of CD34+ cell/kg containing the corrected gene) on examined tissue markers. However, the limited number of patients reduces generalizability and possible confounding factors include initial lesion size and progression rate as well as severity of inflammation. Confirmation of such correlations is warranted as it could inform optimization of the gene drug product...(p.7)

8. In line 299 it is described that also the CD14+ cells have been isolated. It would be of major interest if the CD14+ vector copy numbers per diploid genomes at e.g. 30 days post GT correlates stronger with T2W lesion growth over two years (similar to Fig. 1e) and FA decrease over two years (similar to Fig. 1f).

Following the reviewer's suggestions, we explored possible correlations.

1. There is a strong correlation between VCN in all nucleated cells (VCN) and CD14+ cells (VCN CD14+) at any single time point.

2. But the apparent correlation between applied gene dose (VCN in the gene therapy product [GTP]) and VCN in CD14+ cells at 30 days post treatment initiation disappeared over time likely due to higher variation in the VCNs of CD14+ corrected cells (dilutional effect?)

3. Moreover, no significant correlation between CD14+ VCN 30 days post-transplant and T2 lesion growth or FA decrease at 2 year follow up was found.

We thank the reviewer for asking us to look into these correlations as it revealed a poor correlation between VCN in monocytic cells (CD14+) at the earliest time point (with the lowest variation across subjects and highest VCNs) and lesion progression. Myeloid/monocytic cells have been thought to account for the long-term therapeutic effect after allo-HSCT in ALD by providing an initial large wave of brain engraftment (while the blood brain barrier is disrupted and myeloid precursors in the brain have been depleted by busulfan) and a constant source of functional anti-inflammatory cells with brain turnover of monocytes and microglia.

Our data, demonstrating that VCN in GTP carries the strongest prediction supports the hypothesis of an initial wave of myeloid precursor brain engraftment and less a constant source of circulating monocytic cells (CD14+) as responsible for the long-term effects. In mice assessments at diverse time-points after HSCT showed a short-term wave of brain infiltration by transplanted hematopoietic progenitors mediating local proliferation of early immigrants rather than entrance of mature cells from the circulation. Homing of myeloid precursors to the brain was only possible when the conditioning regimen (i.e. Busulfan) was capable of ablating brain-resident myeloid precursors allowing turnover of microglia from the donor cells (Capotondo et al. 2012).

We have modified the following sentences to reflect these observations:

... The lack of correlation between ABCD1 VCN of CD14+ and growth deceleration (Fig.4c) taken together with the waning correlation with VCN in the drug product over time (suppl.fig.4) suggests that the initial effects could be attributed to a wave of migration and engraftment of ABCD1 expressing circulating progenitor-derived nucleated cells in the peripheral blood (Fig.4d).²² However, the delayed and sustained microvascular flow effects seem to reflect normalized interactions between corrected circulating leukocytes and endothelial cells more likely than, as previously hypothesized, slow turnover of brain resident cells by progeny of engrafted long-term reconstituting HSCs.²³ Further studies are needed to explore if the percentage of transgene-corrected cells impacts imaging markers or clinical outcome or rather certain thresholds need to be met (as suggested by the extreme rarity of CALD in females heterozygotes with one normal copy of the gene).¹⁶...(p.8)

Another possible, still unknown mechanism is the contribution of other cell populations (not CD14+) to the brain microvasculature that could contribute to vessel remodeling and restoration of barrier function. These observations and the known contribution of HSC CD34+ endothelial precursors to an injured blood brain barrier (Garbuzova-Davis et al. 2017, Garbuzova-Davis et al. 2019) led us to investigate the potential contribution of GT and allo-HSCT to brain microvascular engraftment of endothelial cells and pericytes in CALD (see answer to following question and reviewer #2)

9. Discussion line 140: "...1) direct correction of ABCD1-deficient brain microvascular endothelial cells." Does that mean that the authors suspect that the brain microvascular endothelial cells can be exchanged by GT derived cells? The partial reconstitution of brain mutant macrophage/microglial cells and changes in circulating inflammatory cells and their cytokine profiles appear most likely. The correction of monocyte/macrophage lineage by GT would normalize the adherence to microvascular endothelial cells. This would be in the best agreement with the overall long term correction of the even in the NAWM in comparison to the hemizygote X-ALD patient without inflammation. Within the early phase very likely partial reconstitution of brain mutant macrophage/microglial cells as well as the overall change of inflammatory status are of importance for the observed findings. A global direct correction of ABCD1-deficient brain microvascular endothelial cells however would require much more explanation and references in order to be understood.

We sincerely thank the reviewers for the thorough analysis and questions about "...1) direct correction of ABCD1-deficient brain microvascular endothelial cells" as potential mechanism and agree that the correction of circulating myeloid cells and their initial wave of brain engraftment are likely large contributors to the microvascular flow improvements.

However, to determine if correction of bone marrow-derived progenitors could be contributing to functional endothelial precursor cells we studied blood outgrowth endothelial cultures of a subject from our cohort with an ABCD1 mutation known to cause complete loss of ALDP expression (confirmed by flow cytometry of all nucleated cells, including CD34+ and CD14+ peripheral cells) before and after GT. While only a fraction of his CD34+ cells were corrected (23.4% at 24 months) only CD34+ corrected cells (expressing ALDP) gave rise to endothelial precursors and a mature monolayer of characteristic differentiated endothelial cells expressing ALDP in peroxisomes in vitro (BOEC now included as Fig. 4e). This predominance of corrected endothelial cells may arise from advantages to non-corrected precursors with impaired cell cycle regulation and mitochondrial function caused by loss of ABCD1 (Musolino et al. 2012).

1. As stated by the reviewer 1, correction of monocyte/macrophage lineage by GT-corrected myeloid cells likely contribute to the improvement in leukocyte-endothelial interactions and thus improved white matter microvascular flow and barrier functions detected by perfusion MRI (Kapp, CTH, RTH, Vortex Area). Notably, our new human brain histological data following allo-HSCT also demonstrates that direct engraftment of peripheral blood-derived ABCD1-expressing cells to the endothelial and pericyte pool of brain grey and white matter microvessels occurs following myeloid cell correction. The timing and weight of their contribution to microvascular function normalization in ALD remains understudy.

Note that endothelial cells in PBMCs (peripheral blood mononuclear cells) have the highest expression of ABCD1 mRNA:

ABCD1

Source: <https://www.proteinatlas.org/ENSG00000101986-ABCD1/celltype/PBMC>

This is now included in the following paragraph:

Thus, to determine if GT corrected cells could, in addition to myeloid (monocyte/microglial cells), we assessed if CD34⁺-derived peripheral blood cells could give rise to outgrowth endothelial cells (BOEC) before and after GT from a subject with a mutation lacking ALDP expression at baseline.²⁴ We found endothelial precursor colonies and 5 weeks later a mature monolayer of characteristic differentiated endothelial cells expressing ALDP in peroxisomes (Fig.4e). Notably, while only a fraction of leukocytes and monocytes derived from CD34⁺ corrected cells expressed ALDP in this subject at 24 months (12.7% and 23.4% respectively), 100% of surviving differentiated endothelial cells showed ALDP expression. This strongly suggest that uncorrected precursors with impaired cell cycle regulation and

mitochondrial function caused by loss of ABCD1 function^{6,25} are less viable or able to differentiate during in vitro conditions...(p.9)

2. We also confirmed that migration and engraftment into the brain of bone marrow-derived vascular and peri-vascular cells occurs in ALD when we studied a single brain autopsy specimen of a 10 year old boy who succumbed to advance CALD 15 months following peripheral blood engraftment of allo-HSCT. Of note, this is the only subject available with HSCT who survived the acute treatment in the NICHD Brain and Tissue Bank for Developmental Disorders to date. Our immunofluorescence analysis demonstrated not only the characteristic punctate peroxysomal cytoplasmic ALDP expression in circulating and perivascular mononuclear cells (monocytes and macrophages/microglia respectively) but also in endothelium and pericytes of vessels within the white matter surrounding the lesion (perilesional capillaries) but also in its juxtaposed cortex. Localization of ALDP in the peroxisomes was confirmed by co-staining with catalase antibodies. While the specific mutation and level of ALDP expression of the host is unknown, we found no ALDP expressed in oligodendrocytes, resting microglia and astrocytes (known to express ALDP) suggesting a deleterious mutation with no residual ALDP protein.

As the reviewers correctly stated, engrafted cells are likely to concentrate in regions with overt disruption of the BBB as indicated by increased permeability to gadolinium contrast on MRI (Kapp and T1-post contrast enhancement in lesional and perilesional areas in CALD). However, the presence of ALDP expressing pericytes in the cortical microvessels (which have no overt BBB disruption in CALD in MRI or histopathology) suggest engraftment of HSC-derived vascular precursors is possible in areas with relatively normal BBB.

To summarize these findings we added to the text:

While no brain tissue is available in any of the patients treated with GT, we studied a brain autopsy specimen of a boy with CALD who succumbed to advanced disease 15 months following allo-HSCT. This demonstrated for the first time that in a patient with ALD bone marrow-derived cells can engraft long-term in the vascular and perivascular space of white and grey matter. Our immunohistochemistry confirmed that this subject had no detectable levels of ALDP in resident brain cells that normally express the protein endogenously (astrocytes, oligodendrocytes and resting microglia²⁶) but striking characteristic punctate peroxysomal ALDP expression was found in circulating and perivascular monocytic cells, endothelium and pericytes at the demyelinating lesion edge and surrounding cortex (Fig.4f). These data suggest that in addition to correction of the monocyte/macrophage lineage, which could normalize the adherence to microvascular endothelial cells, hematopoietic stem cells could also be providing vascular elements to the cerebral microvasculature that may account for changes in the microvascular function of NAWM on MRI over time (Kapp, CTH, RTH, Vortex Area)...(p.9-10)

Reviewer #2 (Remarks to the Author):

Thank you for the opportunity to review this novel paper from a very able group in this field. Many have been eagerly waiting for more reports of findings related to the GT for this disease. Overall this

paper offers very important new insight into evaluating the pathological processes that occur in cerebral ALD, and the authors provide a critical focus on microvascular function, given what we know about BBB disruption in this devastating disease. I have some questions for the group.

1) I very much appreciate that the authors put the GT findings in context by, at points, comparing to allo-HSCT outcomes. Yet the groups are different by initial lesion severity, with the allo-HSCT group having more severe initial lesions than the GT group. Specifically, while it says that lesion sizes matched between the groups, the supple table 1 suggests otherwise? Initial (i.e., pre-treatment) lesion "size" has been an important predictor of outcomes, including the behavior of the lesion following allo-HSCT. For example, Pierpont and group (Neurology 2020) recently reported that more severe lesions before allo-HSCT tended to increase in severity at a more rapid pace after HSCT; in contrast less severe lesions increased in severity at a less rapid pace after HSCT. This makes me wonder:

a. Given that the allo-HSCT group has more severe initial lesions, which would predict worse growth behavior, should our hopes for GT be muted by the fact that this less severe GT group is showing "Similar lesion growth behavior" (lines 73-74) to a more severe group (allo)? I would appreciate that the authors somehow represent this because to say that the two therapies show similarities might be too reassuring.

We agree with the reviewer that comparison between the allo-HSCT group and the GT group should be made with caution, since there are key differences in the two groups. As also pointed out by reviewer 1, the phrasing lesion size was matched between groups is misleading. We now provided a more detailed description regarding the matching criteria.

In the methods section:

... In addition, male ALD patients with CALD lesion size matching inclusion criteria for rescue treatment (Loes score ≤ 10) and available advanced MRI imaging ... (p.16)

In the main text:

...We investigated lesion growth behavior in a retrospective contemporary cohort treated with allo-HSCT, which also showed lesion growth deceleration following treatment initiation (suppl.fig.1). While this group had similar baseline lesion scores prompting rescue treatment (Loes' score ≤ 10), baseline lesion volumes were significantly larger and observation intervals less frequent (suppl. tabl. 1). This limited our ability to directly compare both treatments... (p.4)

And:

...Overall, we observed similarities between the GT and allo-HSCT groups regarding effects on lesion growth and white matter microcirculation, but the baseline differences in these groups and our design was not sufficient for non-inferiority studies.(p.7)

b. Knowing that pre-treatment lesion severity has been identified as important by other groups, I do not find to be adequate this justification for removing this factor and instead focusing on the behavior of lesions beginning with post-treatment: "To eliminate the effects of baseline lesion variations," as this author group cites their prior work. Perhaps it would reduce the appearance of bias if the authors elaborate upon the scientific rationale for beginning with the lesion severity post treatment and acknowledge others' findings in this regard.

We agree with the reviewer. Lesion size prior to treatment is one of the most significant predictors of treatment response. The cited study Pierpont et al. 2020 used progression in the Loes scoring system as an outcome. We found the Loes scoring system not to capture the lesion trajectory adequately, especially in very early and limited lesions. For instance, smaller lesions have a much higher relative growth rate as compared to larger lesions (up to 46x if the Loes score is 1 vs. >1, Liberato et al. 2019). The main reason for us to compare volumetrics and relative lesion growth rates was to avoid underestimation of lesion trajectory in smaller lesions which are more frequent in the examined GT group compared to the allo-HSCT group. On the other hand, a much higher relative growth rate in small lesions has a similar potential to confound results as the period of prior to treatment and V1, which includes 57.5 days (median, range 10-72) of unhindered lesion progression without applied treatment. In addition, it is reasonable to assume that the immunosuppressive treatment associated with the transplantation has a significant effect in all patients from time of treatment to V1.

The aim of this part of the manuscript was to explore potential gene dose and response correlations. While the data presented in figures 1e and 1f includes lesion growth from baseline, we aimed to exclude the amount of lesion growth that occurs within the period of PRE-V1 in order to have a higher focus on a gene dose mediated effect. We think results of the second approach confirm the firsts. We rephrased to clarify our intentions:

...Baseline lesion variations,¹⁷ the interval before treatment and the immunosuppressive effects of myeloablative conditioning prior to drug product infusion all have potential confounding effects on lesion growth. In order to reduce these effects, we analyzed relative lesion growth after treatment... (p.4-5)

2) Could the authors specify length of time on immunosuppressive treatment? I appreciate that the authors address it and give their assertion that it is unlikely to explain findings. Just wish to know more detail on it.

This information is now provided in short in the main text:

...Our findings that the effect upon BBB-permeability and capillary flow last beyond the period of immunosuppression caused by myeloablative chemotherapy (applied 10-1 days prior to infusion) suggest that immunosuppression and cytokine normalization are not the main mechanism underlying lesion growth deceleration. Further, the effects extend spatially to areas not affected by inflammatory demyelination... (p.9)

And we now include a more detailed description of the conditioning and timing to the methods section of the manuscript:

...Patients received conditioning with busulfan and cyclophosphamide after which the gene therapy product was infused. Weight-based dosing of busulfan IV will be administered on Days -10, -9, -8, and -7 and cyclophosphamide IV (50 mg/kg/day or adjusted dose for obese subjects) will be administered on Days -5, -4, -3, and -2.¹

Patients treated with allo-HSCT received conditioning with busulfan and cyclophosphamide or busulfan and cyclophosphamide (same length of treatment as in GT) with addition of anti-thymocyte globulin or fludarabine if cord blood was the stem-cell source... (p.17)

3) While there are many interesting elements in the figures, they might be a bit busy with so very many panels. Anywhere the authors could be willing to trim? It is leaning toward creating some burden for the reader to sort through the many aspects and map up against the legends and the text.

We appreciate this suggestion. However, to meet all reviewers' requests, we had to add additional data to the figures. In order to improve accessibility, we now group the data into four figures. Also, we moved some graphs to supplementary figures.

4) Possibly rewording for clarity lines 54-55, "Focal increases in these flow disturbances signal where active demyelination will later ensue." Given the wide readership of this journal, I think the less familiar audience may misinterpret this sentence to mean that there are accepted, reliable ways to determine which boys will convert to the cerebral form.

We agree with the reviewer, that to date it has not reliably been shown that CALD patients can be identified prior to conversion. We rephrased this sentence to clarify the important difference between conversion to CALD lesion and lesion progression:

...In addition, regional increases in these flow disturbances precede progression of the active demyelination.^{5,9-11}... (p.3)

Reviewer #3 (Remarks to the Author):

Lauer and the colleague investigated white matter microvascular function based on MRI findings.

The authors described that they analyzed "a sub-cohort of 15 patients". According to the original report (NEJM 2017), 17 patients were enrolled for the gene therapy (GT), and two of them died during the course. The author should explicitly describe this point.

The previously published findings of the ongoing STARBEAM study included 17 Patients. However, the current count of enrolled patients reached 32 at the time the manuscript was crafted. In order to avoid confusion, it needs to be pointed out that the patients investigated in the current study are from the MGH only. None of the patients enrolled at this site died. Investigation of deaths showed that one patient had died from disease progression and another patient, who had had evidence of disease progression on MRI, had to undergo allogeneic stem- cell transplantation and later died from transplantation-related complications. However, to not obscure the fact that GT is an emergency high risk rescue treatment (as would be allo-HSCT) we provide the intention to treat survival rate of the first published study of the STARBEAM results:

...Recently, gene therapy (GT) with autologous hematopoietic stem-cells, which utilizes CD34+ cells transduced *ex vivo* with a lentiviral vector that contains ABCD1-cDNA, has shown promising early results as an alternative to allo-HSCT for children lacking a related donor **with an early 88% survival rate**.¹⁴ ... (p.3)

We now also include some more details in the methods section:

...At the time this manuscript was drafted, 32 patients had been enrolled into the ongoing trial (of which 15 were treated at MGH). Two deaths have occurred since trial initiation (none of the patients treated at MGH have died to this date)... (p.16)

The authors described that Lesion volumetric analysis confirmed that GT effectively arrests CALD disease progression. In the original report (NEJM 2017), lesion progression, as measured with the Loes score, had stabilized in 12 of the 17 patients. The authors need to be careful to describe the results. There seems to be a bias in the description.

We thank the reviewer for pointing this out. We agree that precise phrasing is important. We used volumetric analysis as we found these to be more sensitive to progression, especially in smaller lesions (Liberato et al. 2019). To more adequately reflect our data, we rephrased:

...Lesion volumetric analysis confirmed that GT effectively decelerates CALD lesion progression and is able to halt CALD disease in a large proportion of patients within the observation intervals...(p.4)

The authors described that a rapid deceleration of lesion growth rate occurred as early as 1 month after GT initiation. How can such a conclusive statement be made within one month? The reviewer does not think it is possible.

Our observation is that lesion dynamics showed the most drastic effect of treatment on relative lesion growth (regardless of allo-HSCT or GT) between the PRE and first follow up visit after treatment initiation, which was performed between 26 and 183 days (median: 43.5) in the GT cohort. Our temporal resolution is limited to the predetermined follow up intervals. While we observed this in those patients with a follow up time under 30 days as well, we rephrased this statement to reflect all patients at the first imaging follow up visit:

...The most rapid deceleration of lesion growth rate occurred after the first follow up visit after GT infusion (median days, range; 43.5, 26-183), followed by a less prominent but sustained deceleration over the next two years (Fig.1c)... (p.4)

The authors described that their data support the hypothesis that the correction of ABCD1 expression

in HSC can normalize BBB-permeability and capillary flow dynamics in a dose-dependent manner. The observation of changes in BBB-permeability is interesting. Similar findings have also been described in the patients who underwent HSCT. It is unclear, however, whether ABCD1 expression in HSC directly changed the functions of the endothelial cells. It may well be a secondary result. The authors need careful consideration on this point.

The authors suggest three mechanisms of the effect of GT on BB-permeability, including 1. direct correction of ABCD1-deficient brain microvascular endothelial cells, 2. partial reconstitution of brain mutant macrophage/microglial cells, and/or 3. changes in circulating inflammatory cells and their cytokine profiles. "Endothelial-monocytes interactions" is obscure wording. Consider clear expressions. The authors need to explain the rationale for "direct correction of ABCD1-deficient brain microvascular endothelial cells". Is there any evidence that endothelial cells are directly corrected by GT?

We agree in this with the reviewer. To address this critical point and also the comments of reviewer 1 regarding this section of the manuscript we collected peripheral mononuclear cells (PBMCs) from a patient (with a mutation known to lack ALDP at baseline) before and after GT. We were able to show that only CD34+ corrected cells (expressing ALDP) gave rise to endothelial precursor colonies and later a mature monolayer of characteristic differentiated endothelial cells expressing ALDP in peroxisomes in vitro (BOEC now included as Fig.4e).

These demonstrate that beyond the correction of monocyte/macrophage lineage, GT can also affect elements of the cerebral microvasculature over time. We think these functional vascular and immune system cells account for the differences found in the NAWM of treated subjects (allo-HSCT and GT) when compared to the hemizygote X-ALD subjects without cerebral inflammation (CALD).

We studied brain autopsy specimens of a deceased 10y/o boy with CALD after engraftment of allo-HSCT. Our immunohistochemistry analysis demonstrated the characteristic punctate cytoplasmic ALDP expression in circulating and perivascular mononuclear cells (monocytes and macrophages/microglia respectively) but also in endothelium and pericytes of vessels surrounding the lesion (peri-lesional capillaries) and its juxtaposed cortex.

We have revised our wording as follows:

...Plausible mechanisms by which replacement of HSC-derived cells expressing *ABCD1* could restore endothelial-monocytes interactions and consequently white matter microvascular dysfunction include 1) direct contribution of bone marrow-derived endothelial precursors to turn over of ABCD1-deficient brain microvascular endothelial cells; 2) partial reconstitution of brain mutant macrophage/microglial cells; and/or 3) changes in circulating inflammatory cells and their cytokine profiles... (p.8)

Treatment with the Lenti-D HSC GT halts CALD lesion progression and improves microvascular circulation of the entire cerebral white matter. This description is clearly overstated and should be carefully rephrased. In the original report (NEJM 2017) the clinical effect is limited to partial effect in some patients.

While we do not think we overstated our description of imaging biomarkers, we agree that the impact on clinical outcome is not proven. We now provide additional discussion to reflect this important key difference:

...Taken together, our data support the hypothesis that the correction of *ABCD1* expression in HSC can normalize BBB-permeability and capillary flow dynamics extending beyond the CALD lesion (Fig.4a).

Yet it remains unclear if the observed changes in cerebral white matter microcirculation translate into clinical effects. Functional outcome observed in the STARBEAM study appeared to be worse in patients that received a drug product with lower VCN. Likely the halting of CALD demyelination, cessation of cerebral inflammation and improved metabolic milieu could be contributing as well. (p.7)

It is unclear whether the changes in BBB-permeability are a direct result of changes in endothelial cell functions with GT, or a secondary phenomenon as a result of ABCD1-expressing bone marrow-derived cells. Since this point is essential, the authors need to discuss on this point in detail.

We appreciate this comment as it has added critical value to our manuscript and hope to have addressed it to the best of our capacity in the response to point 9 of reviewer 1.

Recent publication of HSCT for adult CALD (Brain Communications, 2020, <https://doi.org/10.1093/braincomms/fcz048>) should be referred.

We now include this most recent reference in the revised version of the manuscript:

...In early disease stages, allogeneic hematopoietic stem-cell (HSC) transplantation (allo-HSCT) can halt disease progression.¹²⁻¹⁴...(P.3.)

REVIEWER COMMENTS

Reviewer #1 (Remarks to the Author):

The revised version of the manuscript by Lauer et al. did strongly improved. This is not only based on careful rephrasing individual essential passages, but mainly on the exciting results of new experiments that have been added. In particular the new figure 4f is of great interest and substantiate the entire manuscript.

Reviewer #2 (Remarks to the Author):

I commend the authors for meticulously addressing the criticisms and bringing further energy to this investigation by adding in considerable work to clarify and support their conclusions. This paper is vastly improved. Rebuttals are fair and well explained. I have no further concerns.

Reviewer #3 (Remarks to the Author):

1. The most essential issue is the following statement.

Plausible mechanisms by which replacement of HSC-derived cells expressing ABCD1 could restore endothelial-monocytes interactions and consequently white matter microvascular dysfunction include 1) direct contribution of bone marrow-derived endothelial precursors to turn over of ABCD1-deficient brain microvascular endothelial cells; 2) partial reconstitution of brain mutant macrophage/microglial cells; and/or 3) changes in circulating inflammatory cells and their cytokine profiles.

It has been well known that CE disappears first and early after HSCT as well. The critical question is whether the bone marrow-derived endothelial precursors DIRECTLY and CAUSALLY correct the inflammatory demyelination in CALD. The microvascular dysfunction may well be a secondary phenomenon in the inflammatory demyelination. The authors need to be more careful in interpreting the data.

2. The following description needs careful consideration

Gene therapy led to a biphasic response with initial rapid lesional changes (<30-60d) and longer term (>6-12m) more widespread effects extending to NAWM (Fig.4a) and correlating with slower lesion growth (Fig.4b).

Although we observe disappearance of CE in GT and HSCT early, it takes a substantial time to arrest the growth of the lesions (6-24 months) with HSCT and similar response with GT. We are not certain if the disappearance of CE suppresses the entire disease processes in CALD. Again, the authors need to be very careful in this point as well. "slower lesion growth" should also be stressed as disappearance of CE with similar weight.

3. The title, "Gene Correction in Hematopoietic Stem Cells Restores White Matter Microvascular Function in Cerebral Adrenoleukodystrophy" is clearly misleading. "Gene Correction" should be replaced by more appropriate wording. It is not "gene editing".

4. Temporal changes of CE after GT.

The authors described the following sentence; in half of the patients a recurrence of a weak T1-weighted hyperintensity between 6 months and 2 years was reported. This is potentially serious observation, which is rarely observed in HSCT.

5. RTH parameters in NAWM.

The RTH parameter was elevated in white matter of untreated HEM compared to age matched

controls (Fig.3b).

The above description is the RTH parameter of NAWM . How reliable is this measurement? The authors would need to analyze much larger number of patients. Do authors mean that there are significant changes in the RTH parameter even in NAWM of ALD patients?

6. Pathological observation of outgrowth endothelial cells.

Thus, to determine if GT corrected cells could contribute to microvascular cell precursors, in addition to myeloid (monocyte/microglial cells), we assessed if CD34+-derived peripheral blood cells could give rise to outgrowth endothelial cells (BOEC) before and after GT from a subject with a mutation lacking ALDP expression at baseline. We found endothelial precursors colonies and 5 weeks later a mature monolayer of characteristic differentiated endothelial cells expressing ALDP in peroxisomes (Fig.4e).

This paragraph is QUITE confusing. This is a pathological study of a boy with CALD who succumbed to advanced disease 15 months following allo-HSCT. Is this understanding correct? The sentence refers to an autopsied patient treated with HSCT briefly in the later part of the paragraph. It is not even mentioned in the figure legend to Fig. 4.

We would like to thank the reviewers their thoughtful comments and constructive suggestions which have significantly improved our manuscript (NCOMMS-20-38677-T) entitled: "Gene Correction in Hematopoietic Stem Cells Restores White Matter Microvascular Function in Cerebral Adrenoleukodystrophy".

We addressed the remaining reviewers concerns and also the editorial comments requests (detailed response given in the author checklist). Detailed responses to the reviewers' comments are outlined below:

REVIEWER COMMENTS

Reviewer #1 (Remarks to the Author):

The revised version of the manuscript by Lauer et al. did strongly improved. This is not only based on careful rephrasing individual essential passages, but mainly on the exciting results of new experiments that have been added. In particular the new figure 4f is of great interest and substantiate the entire manuscript.

We thank the reviewer for the kind comments on our manuscript.

Reviewer #2 (Remarks to the Author):

I commend the authors for meticulously addressing the criticisms and bringing further energy to this investigation by adding in considerable work to clarify and support their conclusions. This paper is vastly improved. Rebuttals are fair and well explained. I have no further concerns.

We thank the reviewer for the kind comments on our manuscript.

Reviewer #3 (Remarks to the Author):

1. The most essential issue is the following statement.

Plausible mechanisms by which replacement of HSC-derived cells expressing ABCD1 could restore endothelial-monocytes interactions and consequently white matter microvascular dysfunction include 1) direct contribution of bone marrow-derived endothelial precursors to turn over of ABCD1-deficient brain microvascular endothelial cells; 2) partial reconstitution of brain mutant macrophage/microglial cells; and/or 3) changes in circulating inflammatory cells and their cytokine profiles.

It has been well known that CE disappears first and early after HSCT as well. The critical question is whether the bone marrow-derived endothelial precursors DIRECTLY and CAUSALLY correct the inflammatory demyelination in CALD. The microvascular dysfunction may well be a secondary phenomenon in the inflammatory demyelination. The authors need to be more careful in interpreting the data.

We do not infer causality with our statement, on the contrary, we enumerate some of the likely intervening mechanisms and the use of 'plausible mechanisms' is far from probably, possible or proven (causal) and intentionally chosen to allow the reader to look at the data critically and generate new hypothesis. Our current understanding of the disease suggest that CALD is not a cell autonomous disease and therefore we do not expect a single cell (or molecule) to be sufficient to restore white matter homeostasis. We hope we are able to transmit throughout the manuscript is ABCD1 expressing cells (donor cells) can become part of the microvasculature in the human brain after HSCT in CALD (Figure 4f), a finding previously never published. This adds to our previous work demonstrating direct effect of loss of ABCD1 in brain human

microvascular endothelium (*in vitro* modeling) and upon white matter microvascular physiology (*in vivo* patient imaging studies) in ALD (beyond cerebral disease).

We agree with the reviewer that a degree of microvascular dysfunction is secondary to the surrounding inflammatory demyelination. This is seen by the increase of apparent microvascular dysregulation seen within and surrounding inflammatory lesions. However, even in the absence of CALD (overt inflammatory demyelination) loss of ABCD1 function leads to microvascular dysregulation in the human brain. This was demonstrated *in vivo* in human cerebral blood flow measurements in cerebral white matter in hemizygote males (without brain lesions); and by examining *in vitro* ABCD1 silenced human brain microvascular endothelial cells which upregulate adhesion molecules and decrease tight junction proteins, leading to greater adhesion and transmigration of monocytes across the Blood Brain Barrier (see below also response to 5). In addition, previous reports have shown that simple immunosuppression (for examples using steroids, cyclophosphamide, natalizumab) is insufficient to arrest CALD. Loss of gadolinium contrast enhancement (3 months median) is likely due to the delayed treatment responses that go beyond the introduction and rise of ABCD1 sufficient blood based inflammatory cells occur (6 weeks for alloHSCT, 4 weeks for autoHSCT in average) suggesting that other cell types with slower turnover into the brain vascular and perivascular space are involved.

Moreover, progression of lesion growth and persistent contrast enhancement following full myeloablation (highest level of immunosuppression seen in ALD patients) when HSCT engraftment fails argues against the statement of ‘The microvascular dysfunction may well be a secondary phenomenon in the inflammatory demyelination’. If microvascular dysfunction was only secondary to inflammation, myeloablation should be able to stop lesion progression even in the absence of donor cells available for long-term circulation and brain engraftment. To date, one could strongly argue that no biological or small molecule treatment has been able to halt CALD and that a cell therapy is needed to restore and maintained BBB function in this disease. We rephrased the following paragraph and added this important limitation to our study:

...As a limitation the current study does not allow for quantification of the effect of restoration of ABCD1 function in brain endothelium, pericytes or microglia and their relative contributions to inflammation in CALD. Further research is warranted as the identification of key cells/factors and their temporal interaction in this inflammatory process may lead to the development of new treatment approaches and improved monitoring of CALD... (p.11)

2. The following description needs careful consideration

Gene therapy led to a biphasic response with initial rapid lesional changes (<30-60d) and longer term (>6-12m) more widespread effects extending to NAWM (Fig.4a) and correlating with slower lesion growth (Fig.4b).

Although we observe disappearance of CE in GT and HSCT early, it takes a substantial time to arrest the growth of the lesions (6-24 months) with HSCT and similar response with GT. We are not certain if the disappearance of CE suppresses the entire disease processes in CALD. Again, the authors need to be very careful in this point as well. "slower lesion growth" should also be stressed as disappearance of CE with similar weight.

The contrast leakage maps (Kapp) suggest that the BBB is abnormal beyond the "disappearance" of T1 based-CE on clinical imaging (fig.2c). That is why we agree with the reviewer that it is uncertain if CE negative follow up visits can be taken as fully arrested CALD. Fig.4a suggests that BBB restoration extends over a longer period. We emphasized this by rephrasing:

... Gene therapy led to a biphasic response with initial rapid lesional changes (<30-60d) and longer-term (>6-12m) effects including lesional BBB permeability and changes in vascular integrity extending to NAWM (Fig.4a). The latter correlates with slower lesion growth (Fig.4b)...(P.9)

Beyond the explicit text, we expect the volumetric analysis to convey that lesions continue to grow at a slower rate in the first years after treatment.

3. The title, "Gene Correction in Hematopoietic Stem Cells Restores White Matter Microvascular Function in Cerebral Adrenoleukodystrophy" is clearly misleading. "Gene Correction" should be replaced by more appropriate wording. It is not "gene editing".

We appreciate this suggestion, our term correction was aiming at the correction of HSC given by allo or autologous lenti-D gene added HSCs and not gene editing. This terminology will likely cause the same confusion to the readers. To also comply with the editorial suggestion we now changed the title to:

Hematopoietic stem-cell gene therapy is associated with restored white matter microvascular function in cerebral adrenoleukodystrophy (p.1)

In addition, the headline of Figure 2 and the second part of the results was changed as well:

... Effects of allo-HSCT and gene therapy on inflammatory CALD lesions: (p.4 and p.301)

4. Temporal changes of CE after GT.

The authors described the following sentence; in half of the patients a recurrence of a weak T1-weighted hyperintensity between 6 months and 2 years was reported. This is potentially serious observation, which is rarely observed in HSCT.

Yes, re-emerging CE after gene therapy has been described (Eichler et al. 2017 NEJM. PMID: 28976817). It needs to be stated that these findings were based on a dichotomized (yes vs. no) reading and were called positive if the reader determined already a slight increase in SI in post-contrast imaging. In the appendix of the NEJM manuscript we provided examples of the quality of that CE re-appearance to highlight the differences with initial CE. As stated in the third paragraph on page 5, these readings pose several challenges (variability in the timing of image acquisition after contrast injection and the intrinsic decrease in T1-weighted signal intensity caused by tissue structural changes). The very narrow MRI visit intervals are quite unique in this study and the question whether similar observations can be made in patients treated with allo-HSCT is subject of current investigations. The Kapp measurements, a by far less subjective measurement of contrast leakage shows that in both groups GT and allo-HSCT the CALD lesion shows BBB abnormalities even 2Y after treatment (fig. 2c).

5. RTH parameters in NAWM.

The RTH parameter was elevated in white matter of untreated HEM compared to age matched controls (Fig.3b).

The above description is the RTH parameter of NAWM. How reliable is this measurement? The authors would need to analyze much larger number of patients. Do authors mean that there are significant changes in the RTH parameter even in NAWM of ALD patients?

We have previously published findings investigating a different cohort of cerebral X-linked adrenoleukodystrophy patients (Lauer A, et al. Brain. 2017. PMID: 29136088) where indeed abnormal CTH and MTT (both parameters in RTH) and other microvascular flow parameters were statistically significantly different in NAWM and in males without cerebral inflammatory disease. These abnormalities were highest in white matter regions and developmental stages with the highest vulnerability to cerebral disease. See below one of the main figures.

(e) Representative CTH, OEC, CMRO2max and Kapp maps of a 14y old male control (CON) and a 16y old male hemizygote without CALD (HEM) and group comparisons CON vs. HEM. Data are presented

as Tukey plots and are representative of 10 matched CON and 10 HEM cases. * $p < 0.05$, ** $p < 0.01$, two-tailed t-test. (f) Diagram displaying individual WM perfusion mean values and SD (proportional to circle size) in relationship to CTH, mean transit time (MTT) and CMRO₂max in HEM and CON. Deviation from the 45° degree line indicates a mismatch in MTT and CTH.

In that work we analyzed the whole segmented gray and white matter of X-ALD boys without CALD (hemizygotes, HEM) and compared to an age matched control cohort without the ABCD1 dysfunction. Given the relatively small sample size of $n=10$, we calculated an effect size of 0.12 – 0.19 for ABCD-1 deficiency on the DSC-MR perfusion-based imaging marker. While we previously did not investigate RTH, it does strongly correlate with the deviation from the 45° degree line given in figure f (see also Mouridsen et al. JCBFM. PMID: 24938401). This among other findings such as human monocyte-endothelial adhesion assays (Musolino et al. Brain. 2015 PMID: 26377633) let us to thinking that examination of microvascular flow can provide a powerful biomarker in this monogenetic disease. However, as stated in the summarizing paragraph, further studies with larger sample sizes are needed to confirm our findings.

6. Pathological observation of outgrowth endothelial cells.

Thus, to determine if GT corrected cells could contribute to microvascular cell precursors, in addition to myeloid (monocyte/microglial cells), we assessed if CD34+-derived peripheral blood cells could give rise to outgrowth endothelial cells (BOEC) before and after GT from a subject with a mutation lacking ALDP expression at baseline. We found endothelial precursors colonies and 5 weeks later a mature monolayer of characteristic differentiated endothelial cells expressing ALDP in peroxisomes (Fig.4e).

This paragraph is QUITE confusing. This is a pathological study of a boy with CALD who succumbed to advanced disease 15 months following allo-HSCT. Is this understanding correct? The sentence refers to an autopsied patient treated with HSCT briefly in the later part of the paragraph. It is not even mentioned in the figure legend to Fig. 4.

We appreciate this request for clarification. Two different patients were examined. The samples for the *in vitro* blood outgrowth endothelial cell morphology came from a patient treated with GT (Fig.4e). The pathological samples are drawn from an advanced untreated CALD patient and an advanced CALD patient treated with allo-HSCT (Fig.4f). We changed the paragraph, methods and figure legend to highlight the given circumstances:

...Our findings that the effect upon BBB-permeability and capillary flow last beyond the period

of immunosuppression caused by myeloablative chemotherapy (applied 10-1 days prior to

infusion) suggest that immunosuppression and cytokine normalization are not the main

mechanism underlying lesion growth deceleration and may explain why rapid disease

progression occurs when donor HSC failed to engraft despite ablative immunosuppression.¹³

Further, the effects extend spatially to areas not affected by inflammatory demyelination.

To determine if GT associated cells could contribute to microvascular cell precursors, in addition to myeloid (monocyte/microglial cells), we first assessed if CD34+-derived peripheral blood cells could give rise to outgrowth endothelial cells (BOEC) before and after GT from a patient with a mutation lacking ALDP expression at baseline.²⁴ We found endothelial precursors colonies and 5 weeks later a mature monolayer of characteristic differentiated endothelial cells expressing ALDP in peroxisomes (Fig.4e). Notably, while only a fraction of leukocytes and monocytes derived from CD34+ corrected cells expressed ALDP in this patient at 24 months (12.7% and 23.4% respectively), 100% of surviving differentiated endothelial cells showed ALDP expression, strongly suggesting that uncorrected precursors with impaired cell cycle regulation and mitochondrial function caused by loss of *ABCD1* function^{6,25} are less viable or able to differentiate during in vitro conditions.

While no brain tissue is available in any of the patients treated with GT, we then studied a brain autopsy specimen of a boy with CALD who succumbed to advanced disease 15 months following allo-HSCT. This sample demonstrated for the first time that in a patient with ALD bone marrow-derived cells can engraft long-term in the vascular and perivascular space of white and grey matter. Our immunohistochemistry confirmed that this patient had no detectable levels of ALDP in resident brain cells that normally express the protein endogenously (such as astrocytes, oligodendrocytes and resting microglia²⁶) but striking characteristic punctate peroxisomal ALDP expression was found in circulating and perivascular monocytic cells, endothelium and pericytes at the demyelinating lesion edge and also in the surrounding cortex (Fig.4f).... (p.9-10)

And:

...Brain tissue samples from a male CALD patient (10 years of age) who died from advanced disease after receiving allogeneic hematopoietic stem cell transplantation (allo-HSCT) and from an age-matched untreated CALD patient were obtained for analysis... (p.13)

And:

...For blood outgrowth endothelial cell morphology blood from a patient after treatment with GT was collected and separated with a Ficoll density gradient... (p.14)

And:

...e) *In vitro* Blood outgrowth endothelial cell morphology after GT at day 1, 6 and 37 from a patient with at baseline missing ALDP expression. Microphotographs of representative confocal imaging show ALDP expression in peroxisomes as demonstrated by colocalization with catalase... (p.34)